# How to Continually Adapt Text-to-Image Diffusion Models for Flexible Customization?

**Jiahua Dong[1]∗, Wenqi Liang[2]∗, Hongliu Li[3]†, Duzhen Zhang[1]†, Meng Cao[1],
Henghui Ding[4], Salman Khan[1,5], Fahad Shahbaz Khan[1,6]**

[1]Mohamed bin Zayed University of Artificial Intelligence
[2]Shenyang Institute of Automation, Chinese Academy of Sciences
[3]The Hong Kong Polytechnic University  [4]Institute of Big Data, Fudan University
[5]Australian National University  [6]Linköping University
{dongjiahua1995, liangwenqi0123, hongliuli1994, henghui.ding}@gmail.com
{duzhen.zhang, meng.cao, salman.khan, fahad.khan}@mbzuai.ac.ae

## Abstract

Custom diffusion models (CDMs) have attracted widespread attention due to their astonishing generative ability for personalized concepts. However, most existing CDMs unreasonably assume that personalized concepts are fixed and cannot change over time. Moreover, they heavily suffer from catastrophic forgetting and concept neglect on old personalized concepts when continually learning a series of new concepts. To address these challenges, we propose a novel Concept-Incremental text-to-image Diffusion Model (CIDM), which can resolve catastrophic forgetting and concept neglect to learn new customization tasks in a concept-incremental manner. Specifically, to surmount the catastrophic forgetting of old concepts, we develop a concept consolidation loss and an elastic weight aggregation module. They can explore task-specific and task-shared knowledge during training, and aggregate all low-rank weights of old concepts based on their contributions during inference. Moreover, in order to address concept neglect, we devise a context-controllable synthesis strategy that leverages expressive region features and noise estimation to control the contexts of generated images according to user conditions. Experiments validate that our CIDM surpasses existing custom diffusion models. The source codes are available at https://github.com/JiahuaDong/CIFC.

## 1 Introduction

Latent diffusion models (LDMs) [38, 33, 66, 19] have demonstrated unprecedented capabilities in generating high-fidelity images by incorporating large-scale collections of image-text pairs in the latent feature space. Until now, LDMs [4, 30] have achieved remarkable progress in various application fields, including image editing [29, 26], art creation [8, 54], and reconstruction of fMRI brain scans [48]. In order to synthesize some personalized concepts according to user preferences, custom diffusion models (CDMs) [40, 60, 11] rely on low-rank adaptation (LoRA) [14] to finetune the large-scale LDMs for multi-concept customization [22]. They extend the vision-language dictionary of pretrained LDMs to bind personalized concepts with specific subjects users need to synthesize.

Generally, most existing CDMs [60, 68, 3] assume that users' personalized concepts are fixed and cannot incrementally increase over time. However, this assumption is unrealistic in real-world applications, where users want to continually synthesize a series of new personalized concepts from their own lives. To address this setting, CDMs [3, 57, 2] typically require storing all image-text

---

∗Equal contributions (ordered alphabetically).  †Corresponding authors.

training pairs of old concepts to finetune the pretrained LDMs via LoRA [14, 59]. Nevertheless, the high computation costs and privacy concerns [52] may render CDMs impractical as the number of old personalized concepts consecutively increases. If the above CDMs retain all low-rank weights associated with old concepts that are obtained in previous customization tasks and then merge them to learn new personalized concepts continually [59, 65], they may experience significant loss of individual attributes on old personalized concepts (*i.e.*, catastrophic forgetting [36, 6]) for versatile customization. Moreover, in real-world scenarios, users may wish to control the contexts and objects associated with multiple old concepts in synthesized images according to the conditions they provide (*e.g.*, scribble or bounding box [24, 44]). It forces CDMs [60] to heavily suffer from the challenge of concept neglect [1] (*i.e.*, some old concepts are missing during multi-concept composition).

To handle the above real-world scenarios, in this paper, we propose a new practical problem named *Concept-Incremental Flexible Customization (CIFC)*. In the CIFC setting, as shown in Fig. 1(a), CDMs can consecutively synthesize a sequence of new personalized concepts in a concept-incremental manner for versatile customization (*e.g.*, multi-concept generation [21], style transfer [56] and image editing [29]). Additionally, users can control the context and objects of the generated images based on the specific conditions they provide. As aforementioned, the CIFC problem faces two main challenges for versatile concept customization in this paper: **catastrophic forgetting** of old personalized concepts when learning new concepts consecutively under a concept-incremental manner, and **concept neglect** when performing multi-concept composition according to users-provided conditions.

To resolve the challenges in CIFC, we develop a novel Concept-Incremental text-to-image Diffusion Model (CIDM), which can effectively address catastrophic forgetting and concept neglect. **On one hand**, to mitigate the catastrophic forgetting of old personalized concepts, we propose a novel concept consolidation loss for training and devise an elastic weight aggregation (EWA) module for inference. This loss employs learnable layer-wise concept tokens and an orthogonal subspace regularizer to explore task-specific knowledge (*i.e.*, unique attributes of personalized concepts), while learning layer-wise common subspaces across different tasks to capture task-shared knowledge. Additionally, the EWA module utilizes learnable layer-wise concept tokens to merge all low-rank weights of old personalized concepts, based on their contributions to versatile concept customization. **On the other hand**, we develop a context-controllable synthesis strategy to tackle concept neglect for multi-concept composition. It leverages layer-wise textual embeddings to enhance the expressive ability of region features and relies on region noise estimation to control the contexts of generated image, conforming to users-provided conditions. Comprehensive experiments illustrate the effectiveness of our proposed CIDM in addressing the CIFC problem. The main contributions of this paper are listed below:

• We propose a new practical problem named Concept-Incremental Flexible Customization (CIFC), where the main challenges are catastrophic forgetting and concept neglect. To address the challenges in the CIFC problem, we develop a novel Concept-Incremental text-to-image Diffusion Model (CIDM), which can learn new personalized concepts continuously for versatile concept customization.

• We devise a concept consolidation loss and an elastic weight aggregation module to mitigate the catastrophic forgetting of old personalized concepts, by exploring task-specific/task-shared knowledge and aggregating all low-rank weights of old concepts based on their contributions in the CIFC.

• We develop a context-controllable synthesis strategy to tackle concept neglect. This strategy controls the contexts of synthesized images according to user conditions by enhancing the expressive ability of region features with layer-wise textual embeddings and incorporating region noise estimation.

## 2    Related Work

**Incremental Learning** [25, 49, 20] accumulates previous experience to incrementally learn new tasks without the need for retraining from scratch. To prevent catastrophic forgetting of old tasks, most incremental or continual learning models mainly employ knowledge distillation between old and new tasks [63, 36, 5], replay some images from old tasks [16, 43], or dynamically expand network architecture to encode new knowledge [61, 15, 7]. Nevertheless, these methods [16, 61, 49, 36] are primarily designed for classifying new object categories consecutively, which cannot be directly applied to tackle continual concept customization tasks without catastrophic forgetting.

**Concept Customization** [11, 40, 22] focuses on extending large-scale diffusion model [31, 46, 64] to synthesize personalized concepts for users. After [40] proposes to tackle the subject-driven generation

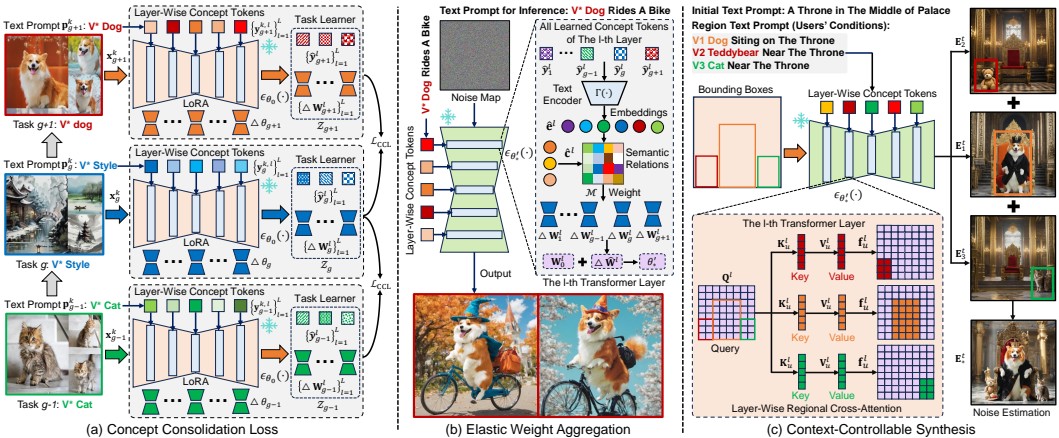

Figure 1: Diagram of the proposed CIDM to address the CIFC problem. It consists of (a) a concept consolidation loss, (b) an elastic weight aggregation module to resolve catastrophic forgetting, and (c) a context-controllable synthesis strategy to address the challenge of concept neglect.

by finetuning all network parameters of the pretrained diffusion model [48] on personalized concepts, some works use textual inversion [9, 53] to learn word embeddings of personalized concepts [50]. For multi-concept customization, [22] can jointly train multiple concepts or combine different diffusion models by optimizing a few parameters in the cross-attention layers, while [27] aims to capture different clusters of concept neurons. Motivated by [22], Han *et al.* [12] finetune the singular values of latent encoding weights, thereby improving the efficiency for concept customization. [42, 57, 18] perform efficient test-time customization by training concept-specific encoders. Besides, [59, 11] fuse multiple low-rank weights to resolve multi-concept customization. In order to tackle continual text-to-image synthesis tasks, Sun *et al.* [47] devise a lifelong diffusion model to accumulate concept information. Unfortunately, it cannot control the contexts of synthesized images and suffers from concept neglect in multi-concept composition [62, 58]. To address the issue of missing concepts, [66, 30] utilize spatial conditions (*e.g.*, sketch and pose) for composition. However, these custom diffusion models [59, 47, 51, 58] cannot consecutively learn a sequence of new concepts to tackle the CIFC problem, as they face challenges related to catastrophic forgetting and concept neglect.

## 3    Preliminary and Problem Definition

**Preliminary:** Latent diffusion models (LDMs) [41, 10] rely on some conditional inputs (*e.g.*, text prompt [33, 37] or image [17, 32]) to control the contexts of synthesized images. They use an encoder $\mathcal{E}(\cdot)$ and a decoder $\mathcal{D}(\cdot)$ to perform image synthesis in the latent space [33]. Custom diffusion models (CDMs) [12, 9, 28] utilize low-rank adaptation (LoRA) [14, 55] to learn new personalized concepts by finetuning the pretrained LDMs [1, 38]. Given a pair of personalized image $\mathbf{x}$ and its text prompt $\mathbf{p}$, the encoder $\mathcal{E}(\cdot)$ maps $\mathbf{x}$ to a latent feature $\mathbf{z}$, and $\mathbf{z}_t$ denotes the noisy latent feature at the $t$-th ($t = 1, \cdots, T$) timestep. After the text encoder $\Gamma(\cdot)$ (*e.g.*, pretrained CLIP [35]) maps $\mathbf{p}$ to the textual embedding $\mathbf{c} = \Gamma(\mathbf{p})$, the objective to learn personalized concept $\{\mathbf{x}, \mathbf{p}\}$ at the $t$-th timestep is:

$$\mathcal{L}_{\text{CDMs}} = \mathbb{E}_{\mathbf{z} \sim \mathcal{E}(\mathbf{x}), \mathbf{c}, \epsilon \sim \mathcal{N}(0, \mathbf{I}), t}[\|\epsilon - \epsilon_{\theta'}(\mathbf{z}_t | \mathbf{c}, t)\|_2^2], \tag{1}$$

where $\epsilon_{\theta'}(\cdot)$ denotes the denoising UNet proposed in [38, 33], and it can gradually denoise $\mathbf{z}_t$ by predicting the noisy estimation $\epsilon_{\theta'}(\mathbf{z}_t | \mathbf{c}, t)$ as Gaussian noise $\epsilon \sim \mathcal{N}(0, \mathbf{I})$. In the paper, $\theta' = \theta_0 + \triangle\theta$ consists of the pretrained parameter $\theta_0 = \{\mathbf{W}_0^l\}_{l=1}^L$ in LDMs [39, 33] and low-rank parameter $\triangle\theta = \{\triangle\mathbf{W}^l\}_{l=1}^L$ updated by LoRA [22, 11]. $\mathbf{W}_0^l, \triangle\mathbf{W}^l \in \mathbb{R}^{a \times b}$ denote the pretrained weight and low-rank weight in the $l$-th ($l = 1, \cdots, L$) transformer layer of $\theta'$, respectively. $a$ and $b$ are the row and column of matrices. As introduced in [40, 68], $\triangle\mathbf{W}^l = \mathbf{A}^l\mathbf{B}^l$ can be decomposed as two low-rank factors $\mathbf{A}^l \in \mathbb{R}^{a \times r}$ and $\mathbf{B}^l \in \mathbb{R}^{r \times b}$, where $r \ll \min(a, b)$ denotes the rank.

However, most CDMs [11, 12, 9, 28] assume that the number of users' personalized concepts remains constant over time. This assumption is unrealistic in real-world applications, where users wish to consecutively generate a series of new personalized concepts based on their preferences. More

importantly, they significantly suffer from catastrophic forgetting [36] of old personalized concepts and concept neglect when performing versatile customization in a concept-incremental manner.

**Problem Definition:** To address the above challenges, we propose a new practical problem named Concept-Incremental Flexible Customization (CIFC). In the CIFC, there is a series of consecutive text-guided concept customization tasks $\mathcal{T} = \{\mathcal{T}_g\}_{g=1}^G$, where $G$ denotes the task quantity. According to users' personal preferences, the $g$-th task $\mathcal{T}_g = \{\mathbf{x}_g^k, \mathbf{p}_g^k, \mathbf{y}_g^k\}_{k=1}^{n_g}$, which includes $n_g$ triplets of image $\mathbf{x}_g^k$, text prompt $\mathbf{p}_g^k$, and its concept tokens $\mathbf{y}_g^k \in \mathcal{Y}_g$, belongs to one of the versatile customization categories: multi-concept generation [69], style transfer [67] and image editing [3]. Here, $\mathbf{p}_g^k$ indicates the textual description of $\mathbf{x}_g^k$ (*e.g.*, photo of a $[V_*]$ $[V_{\text{dog}}]$), whereas $\mathbf{y}_g^k$ denotes the concept tokens (*e.g.*, $[V_*]$ $[V_{\text{dog}}]$) in $\mathbf{p}_g^k$. $\mathcal{Y}_g$ is concept space of the $g$-th task, and it comprises $C_g$ new personalized concepts $\mathbf{y}_g = \cup_{k=1}^{n_g} \mathbf{y}_g^k$ in the $g$-th task. Particularly, the concept spaces between any two tasks have no overlap: $\mathcal{Y}_g \cap (\cup_{i=1}^{g-1} \mathcal{Y}_i) = \emptyset$. It implies that $C_g$ new concepts in the $g$-th task are different from $\sum_{i=1}^{g-1} C_i$ old personalized concepts from $g-1$ old tasks under the CIFC setting. Considering the practicality of the CIFC setting, we don't allocate any memory storage to store or replay the training data of all tasks $\{\mathcal{T}_g\}_{g=1}^G$, ensuring that all concept customization tasks are learned in a concept-incremental manner. The CIFC setting can continually learn new personalized concepts in a concept-incremental manner for versatile customization while tackling the forgetting on old concepts.

## 4 The Proposed Model

Fig. 1 shows the diagram of our concept-incremental diffusion model (CIDM) to tackle the CIFC problem. It includes (a) a concept consolidation loss in Sec. 4.1 and (b) an elastic weight aggregation module in Sec. 4.2 to resolve catastrophic forgetting during training and inference. Additionally, it encompasses (c) a context-controllable synthesis strategy in Sec. 4.3 to address concept neglect.

### 4.1 Concept Consolidation Loss

In order to learn the $g$-th text-guided concept customization task $\mathcal{T}_g$, we use the LoRA [14, 38] to finetune the pretrained denoising UNet $\epsilon_{\theta_0}(\cdot)$ on personalized data $\{\mathbf{x}_g^k, \mathbf{p}_g^k, \mathbf{y}_g^k\}_{k=1}^{n_g}$ by optimizing Eq. (1), and then obtain an updated model $\epsilon_{\theta_g'}(\cdot)$, where $\theta_g' = \theta_0 + \triangle\theta_g$, $\triangle\theta_g = \{\triangle\mathbf{W}_g^l\}_{l=1}^L$, and $\triangle\mathbf{W}_g^l = \mathbf{A}_g^l \mathbf{B}_g^l \in \mathbb{R}^{a \times b}$ is the updated low-rank weight in the $l$-th layer. $\mathbf{A}_g^l \in \mathbb{R}^{a \times r}$ and $\mathbf{B}_g^l \in \mathbb{R}^{r \times b}$ are low-rank factors. As introduced in [14, 60], $\triangle\theta_g$ can encode most of $C_g$ personalized concept identity within the $g$-th task. To address the CIFC problem, a trivial solution for learning the $g$-th task is to store the updated low-rank weights of all tasks $\{\mathcal{T}_i\}_{i=1}^g$ learned so far, and then linearly merge them by evaluating their contributions [59, 65] during training. However, it may result in substantial loss of individual characteristics within some personalized concepts, when learning new concepts continually in the CIFC setting. This phenomenon is referred to as catastrophic forgetting on old personalized concepts. To mitigate catastrophic forgetting during training, as show in Fig. 1(a), we develop a concept consolidation loss $\mathcal{L}_{\text{CCL}}$ to explore task-specific and task-shared knowledge.

**Task-Specific Knowledge** indicates distinctive characteristics of personalized concepts within each concept customization task. To explore this knowledge, we introduce learnable layer-wise concept tokens to better preserve unique attributes of personalized concepts in the synthesized images. It is significantly different from existing textual inversion methods [9, 67, 53] that inject a unified text prompt into all transformer layers of $\epsilon_{\theta_g'}(\cdot)$. For a triplet $\{\mathbf{x}_g^k, \mathbf{p}_g^k, \mathbf{y}_g^k\}$ in the $g$-th task, we define $L$ layer-wise text prompts as $\{\mathbf{p}_g^{k,l}\}_{l=1}^L$, where $\mathbf{p}_g^{k,l}$ has its own learnable layer-wise concept tokens $\mathbf{y}_g^{k,l}$ in the $l$-th transformer layer. For example, given a textual description $\mathbf{p}_g^k$ (photo of a $[V_*]$ $[V_{\text{dog}}]$) of image $\mathbf{x}_g^k$, its text prompt of the $l$-th layer is defined as $\mathbf{p}_g^{k,l}$ (photo of a $[V_*^l]$ $[V_{\text{dog}}^l]$), and $[V_*^l]$ $[V_{\text{dog}}^l]$ indicates learnable concept tokens $\mathbf{y}_g^{k,l}$ in the $l$-th layer. After using $\mathbf{p}_g^k$ to initialize $\{\mathbf{p}_g^{k,l}\}_{l=1}^L$, we inject the textual embedding $\mathbf{c}_g^{k,l} = \Gamma(\mathbf{p}_g^{k,l})$ encoded via the text encoder $\Gamma(\cdot)$ into the $l$-th layer of $\epsilon_{\theta_g'}(\cdot)$. When we train $\epsilon_{\theta_g'}(\cdot)$ via Eq. (1), the learnable layer-wise concept tokens can capture unique characteristics of old personalized concepts from different layers to surmount catastrophic forgetting.

However, the discriminative ability of task-specific knowledge to distinguish different personalized concepts can significantly deteriorate as the number of concept customization tasks gradually increases

under the CIFC settings. To tackle this issue, we devise an orthogonal subspace regularizer to constrain the low-rank weights of different customization tasks. It can enhance the discriminative ability of task-specific knowledge by ensuring the orthogonality of concept subspaces across different tasks. Given the low-rank weight $\triangle\theta_g = \{\triangle\mathbf{W}_g^l\}_{l=1}^L$ in the $g$-th task, $\triangle\mathbf{W}_g^l = \mathbf{A}_g^l\mathbf{B}_g^l$ can be regarded as consisting of the low-rank concept subspace $\mathbf{A}_g^l = [\mathbf{a}_g^{l,1}, \cdots, \mathbf{a}_g^{l,r}] \in \mathbb{R}^{a \times r}$ and its linear weighting matrix $\mathbf{B}_g^l = [\mathbf{b}_g^{l,1}, \cdots, \mathbf{b}_g^{l,r}]^\top \in \mathbb{R}^{r \times b}$, where $\mathbf{b}_g^{l,i} \in \mathbb{R}^b$ denotes the linear weighting coefficients of $\mathbf{a}_g^{l,i} \in \mathbb{R}^a$ ($i = 1, \cdots, r$). In the $g$-th task, we perform the orthogonal subspace regularizer on the low-rank concept subspaces of different tasks: $\sum_{i=1}^{g-1}\sum_{l=1}^L \mathbf{A}_i^l(\mathbf{A}_g^l)^\top = 0$. Since the orthogonal constraint is not differentiable, we propose an alternative optimization strategy that minimizes the absolute value of the inner product between different subspaces: $\mathcal{R}_1 = \sum_{i=1}^{g-1}\sum_{l=1}^L \mathbf{A}_i^l(\mathbf{A}_g^l)^\top$.

**Task-Shared Knowledge** represents the shared semantic information across different tasks with semantically similar concepts, which is beneficial to address catastrophic forgetting on old personalized concepts. To capture task-shared knowledge, we propose to learn a layer-wise common subspace $\mathbf{W}_*^l \in \mathbb{R}^{a \times b}$ shared across different tasks in the $l$-th layer. Given the low-rank weights $\{\triangle\theta_i\}_{i=1}^g$ learned so far, the learnable projection matrix $\mathbf{H}_i^l \in \mathbb{R}^{a \times a}$ can encode common semantic information of $\{\triangle\theta_i\}_{i=1}^g$ into $\mathbf{W}_*^l$ via $\mathcal{R}_2 = \sum_i^g\sum_l^L \|\triangle\mathbf{W}_i^l - \mathbf{H}_i^l\mathbf{W}_*^l\|_F^2$. Therefore, in the $g$-th task, the concept consolidation loss $\mathcal{L}_{\text{CCL}}$ to learn both task-specific and task-shared knowledge is defined as:

$$\mathcal{L}_{\text{CCL}} = \mathbb{E}_{\mathbf{z} \sim \mathcal{E}(\mathbf{x}_g^k), \mathbf{c}_g^k, \epsilon \sim \mathcal{N}(0, \mathbf{I}), t}[\|\epsilon - \epsilon_{\theta_g'}(\mathbf{z}_t|\mathbf{c}_g^k, t)\|_2^2 + \gamma_1\mathcal{R}_1 + \gamma_2\mathcal{R}_2], \tag{2}$$

where $\mathbf{c}_g^k = \{\mathbf{c}_g^{k,l}\}_{l=1}^L$ indicates $L$ layer-wise textual embeddings, $\gamma_1$ and $\gamma_2$ are balance parameters.

**Two-Step Optimization:** To train the proposed CIDM via Eq. (2) in the $g$-th task, we devise a two-step optimization strategy in each training batch. Firstly, to capture task-specific information, we utilize Eq. (2) to update the learnable layer-wise concept tokens and low-rank weight $\triangle\theta_g = \{\triangle\mathbf{W}_g^l\}_{l=1}^L$, when fixing $\mathbf{H}_i^l$ and $\mathbf{W}_*^l$. Secondly, to explore task-shared knowledge, we fix $\triangle\theta_g$ and the learnable layer-wise concept tokens, and then only use $\mathcal{R}_2$ in Eq. (2) to update $\mathbf{H}_i^l$ and $\mathbf{W}_*^l$ respectively. Notably, the detailed optimization procedure is shown in the appendix section (Sec. A.1). After utilizing the two-step optimization strategy to learn the $g$-th concept customization task, we can obtain the task learner $\mathcal{Z}_g = \{\triangle\mathbf{W}_g^l, \widehat{\mathbf{y}}_g^l\}_{l=1}^L$, where $\widehat{\mathbf{y}}_g^l = \{\widehat{\mathbf{y}}_g^{l,i}\}_{i=1}^{C_g}$, and $\widehat{\mathbf{y}}_g^{l,i}$ is the $i$-th ($i = 1, \cdots, C_g$) concept token learned at the $l$-th transformer layer through Eq. (2).

### 4.2 Elastic Weight Aggregation

To tackle catastrophic forgetting on old personalized concepts during inference, as shown in Fig. 1(b), we store $g$ task learners $\{\mathcal{Z}_i\}_{i=1}^g$ learned so far ($g \geq 2$), and develop an elastic weight aggregation (EWA) module to adaptively merge them for versatile concept customization. Note that the memory storage of storing $\mathcal{Z}_i$ ($i = 1, \cdots, g$) only accounts for $0.25\%$ of the pretrained model $\theta_0$, making it negligible in practical applications. Specifically, given a text prompt $\widehat{\mathbf{p}}$ for inference, we use it to initialize layer-wise text prompts $\{\widehat{\mathbf{p}}^l\}_{l=1}^L$ and extract their textual embeddings $\widehat{\mathbf{c}} = \{\widehat{\mathbf{c}}^l \in \mathbb{R}^{n_e \times d}\}_{l=1}^L$ via text encoder $\Gamma(\cdot)$, where $n_e$ and $d$ denote the token number and feature dimension, respectively. Given the stored task learners $\{\mathcal{Z}_i\}_{i=1}^g$, we can collect all concept tokens $\{\Phi^l\}_{l=1}^L$ learned so far, where $\Phi^l = \cup_{i=1}^g\widehat{\mathbf{y}}_i^l \in \mathbb{R}^{n_c}$ consists of $n_c = \sum_{i=1}^g C_i$ concept tokens learned in the $l$-th layer. After $\Gamma(\cdot)$ encodes $\{\Phi^l\}_{l=1}^L$ to latent embeddings $\{\mathbf{e}^l \in \mathbb{R}^{n_c \times d}\}_{l=1}^L$, we average those latent embeddings belonging to the same task to obtain new embeddings $\{\widehat{\mathbf{e}}^l \in \mathbb{R}^{g \times d}\}_{l=1}^L$ ($g$ is number of tasks learned so far). Subsequently, we compute semantic relations $\mathcal{M} \in \mathbb{R}^g$ between $\widehat{\mathbf{c}}^l$ and $\widehat{\mathbf{e}}^l$, and use $\mathcal{M}$ to adaptively merge low-rank weights $\{\triangle\mathbf{W}_i^l\}_{i=1}^g$ of all learned tasks in the $l$-th layer. Therefore, the merged low-rank weight $\triangle\widehat{\mathbf{W}}^l$ in the $l$-th transformer layer can be formulated as follows:

$$\mathcal{M} = \max\left(\widehat{\mathbf{c}}^l \cdot (\widehat{\mathbf{e}}^l)^\top\right), \quad \triangle\widehat{\mathbf{W}}^l = \sum_{i=1}^g \triangle\mathbf{W}_i^l \cdot \psi(\mathcal{M})_i, \tag{3}$$

where $\max(\cdot)$ denotes the maximization function along the row axis. $\psi(\mathcal{M}) = \mathcal{M}^2/\|\mathcal{M}^2\|_F \in \mathbb{R}^g$ is used to normalize the semantic relations $\mathcal{M}$, and $\psi(\mathcal{M})_i$ is the $i$-th element of $\psi(\mathcal{M})$.

**Inference:** After employing Eq. (3) to merge all low-rank weights learned so far, we obtain a new denoising UNet $\epsilon_{\theta_*'}(\cdot)$ for inference, where $\theta_*' = \theta_0 + \triangle\theta_*$, and $\triangle\theta_* = \{\triangle\widehat{\mathbf{W}}^l\}_{l=1}^L$. Notably, $\theta_*'$ has encoded substantial distinctive attributes of all personalized concepts learned so far, which can effectively mitigate the catastrophic forgetting of old concepts during inference.

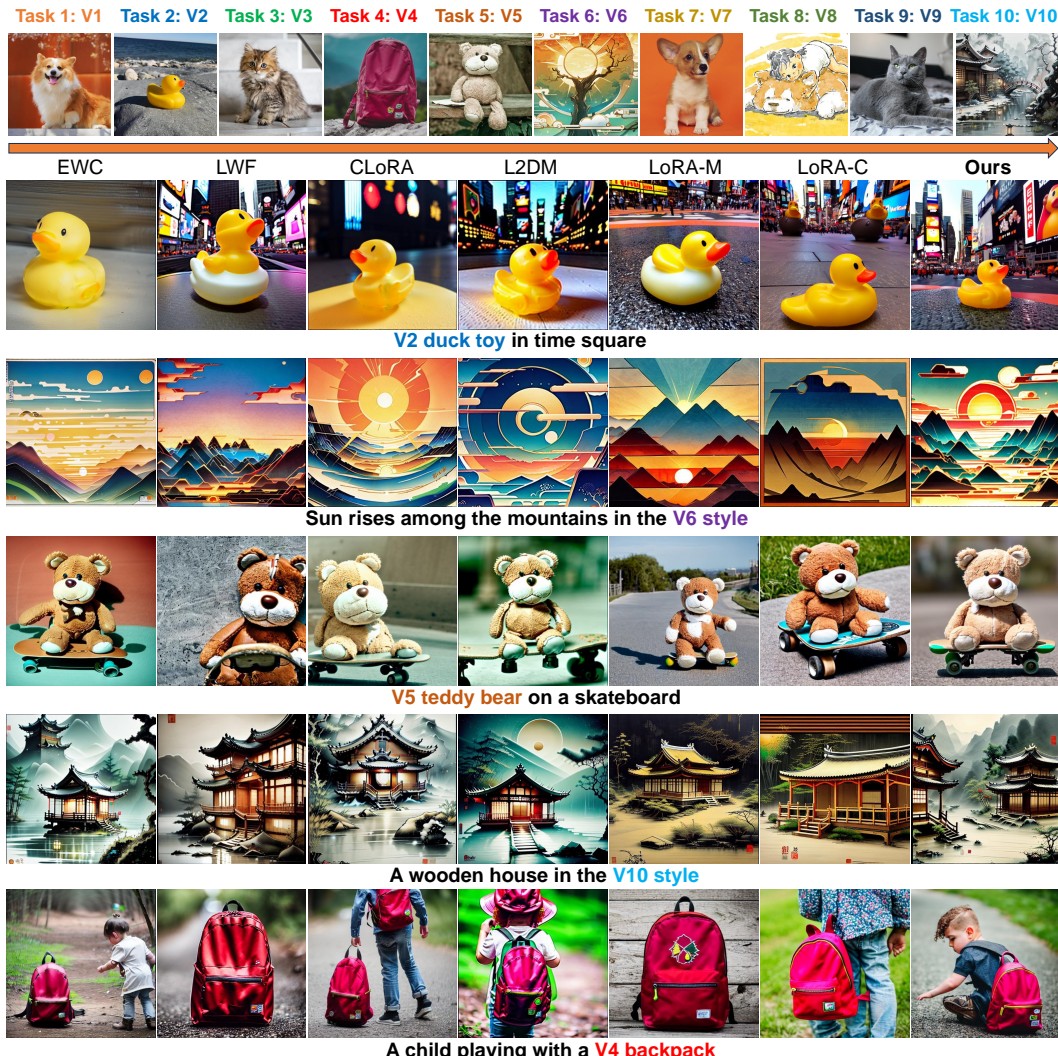

Figure 2: Some qualitative comparisons of single-concept customization generated by SD-1.5 [38].

## 4.3 Context-Controllable Synthesis

When we directly use $\epsilon_{\theta'_*}(\cdot)$ obtained via Sec. 4.2 to perform multi-concept customization under the CIFC setting, it cannot generate high-fidelity images according to users-provided conditions (*e.g.*, scribble or bounding box [24]), and heavily suffers from the challenge of concept neglect [1] (*i.e.*, some concepts are missing in the synthesized images). Thus, as shown in Fig. 1(c), we devise a context-controllable synthesis strategy to address the conditional generation and concept neglect.

**Conditional Generation:** Besides the initial text prompt $\widehat{\mathbf{p}}$ defined in Sec. 4.2, users can provide $U$ pairs of region conditions $\{\widehat{\mathbf{p}}_u, \widehat{\mathbf{s}}_u\}_{u=1}^U$, where $\widehat{\mathbf{s}}_u$ is the bounding box to synthesize concepts related to the $u$-th region text prompt $\widehat{\mathbf{p}}_u$. Then we use $\Gamma(\cdot)$ to extract layer-wise textual embeddings $\widehat{\mathbf{c}}_u = \{\widehat{\mathbf{c}}_u^l \in \mathbb{R}^{n_e \times d}\}_{l=1}^L$ for $\widehat{\mathbf{p}}_u$. Given the initial text prompt $\widehat{\mathbf{p}}$, we can use the new denoising UNet $\epsilon_{\theta'_*}(\cdot)$ in Sec. 4.2 to obtain its feature map $\mathbf{f}^l \in \mathbb{R}^{h^l \times w^l \times d}$ in the $l$-th transformer layer encoded by textual embedding $\widehat{\mathbf{c}}^l$, where $h^l$ and $w^l$ are height and width of $\mathbf{f}^l$. Different from [22], we perform layer-wise regional cross-attention between textual embedding $\widehat{\mathbf{c}}_u^l$ and $\mathbf{f}^l$ to obtain the $u$-th region feature $\mathbf{f}_u^l \in \mathbb{R}^{h_u^l \times w_u^l \times d}$ in the $l$-th layer, where $h_u^l$ and $w_u^l$ are height and width of bounding box $\widehat{\mathbf{s}}_u$. Specifically, $\mathbf{f}_u^l = \sigma(\mathbf{Q}^l(\mathbf{K}_u^l)^\top / \sqrt{d}) \cdot \mathbf{V}_u^l$, where $\sigma(\cdot)$ is sigmoid function, $\mathbf{Q}^l = \Omega(\mathbf{f}^l \mathbf{w}_q \odot \widehat{\mathbf{m}}_u^l) \in \mathbb{R}^{h_u^l \times w_u^l \times d}$, $\mathbf{K}_u^l = \widehat{\mathbf{c}}_u^l \mathbf{w}_k \in \mathbb{R}^{n_e \times d}$ and $\mathbf{V}_u^l = \widehat{\mathbf{c}}_u^l \mathbf{w}_v \in \mathbb{R}^{n_e \times d}$. $\widehat{\mathbf{m}}_u^l \in \mathbb{R}^{h^l \times w^l}$ is the binary region mask in the $l$-th layer, where the values inside the bounding box $\widehat{\mathbf{s}}_u$ are set to 1. $\Omega(\cdot)$ can retain only

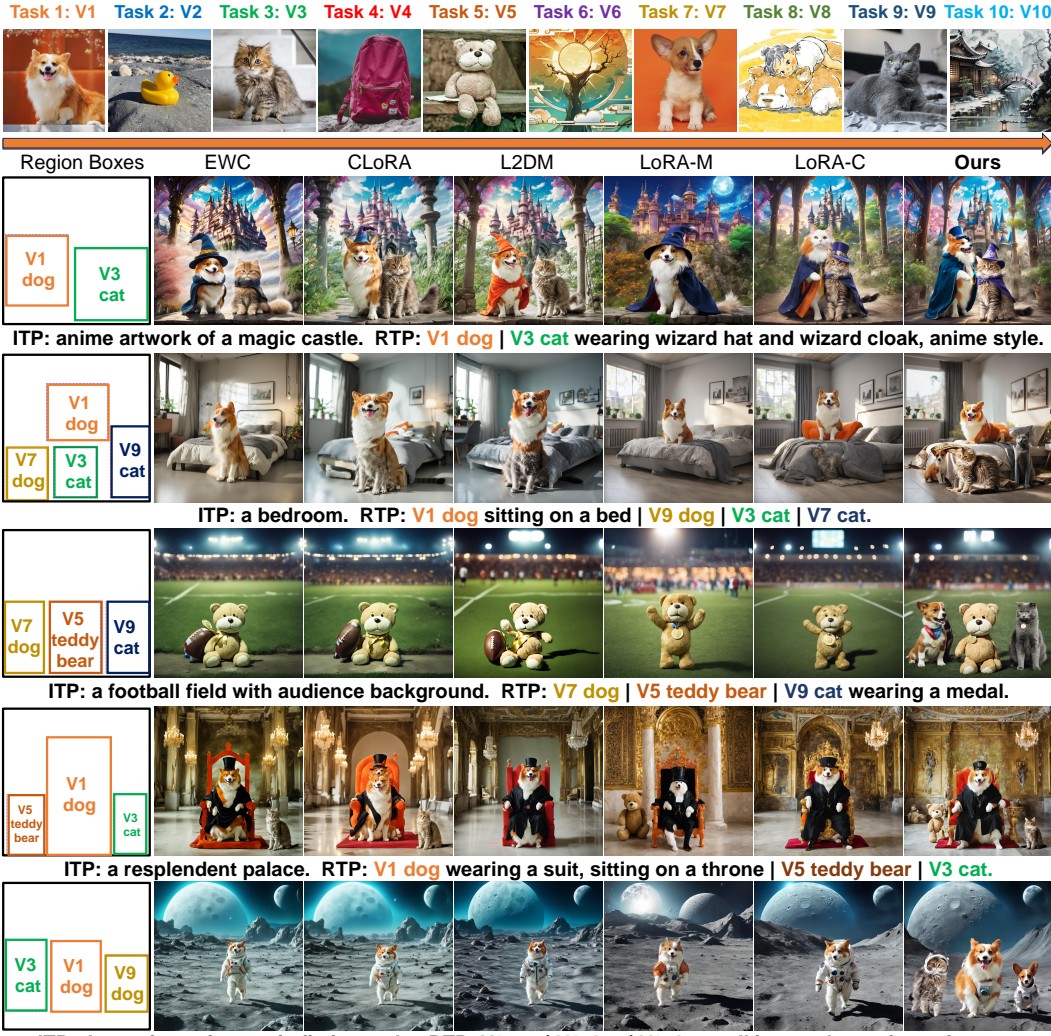

Figure 3: Some qualitative comparisons of multi-concept customization generated by SDXL [33], where ITP indicates the initial text prompt, and RTP denotes the region text prompt.

the features inside $\widehat{\mathbf{s}}_u$, and $\odot$ is the Hardmard product. $\mathbf{w}_q, \mathbf{w}_k, \mathbf{w}_v \in \mathbb{R}^{d \times d}$ are mapping matrices in the new denoising UNet $\epsilon_{\theta'_*}(\cdot)$. Then, the values of $\mathbf{f}^l$ inside the bounding boxes $\{\widehat{\mathbf{s}}_u\}_{u=1}^U$ are respectively replaced with the corresponding region features $\{\mathbf{f}^l_u\}_{u=1}^U$ to obtain a new feature map $\widehat{\mathbf{f}}^l$. We apply the layer-wise regional cross-attention to all layers of $\epsilon_{\theta'_*}(\cdot)$ for conditional generation.

**Multi-Concept Composition:** After integrating the above conditional generation into the new denoising UNet $\epsilon_{\theta'_*}(\cdot)$, inspired by [13], we can obtain the noise estimation $\mathbf{E}^t = \epsilon_{\theta'_*}(\mathbf{z}_t|t) + s \cdot (\epsilon_{\theta'_*}(\mathbf{z}_t|\widehat{\mathbf{c}}, t) - \epsilon_{\theta'_*}(\mathbf{z}_t|t)) \in \mathbb{R}^{h_L \times w_L \times d_L}$ for the initial text prompt $\widehat{\mathbf{p}}$, where $\mathbf{E}^t$ is the output of the $L$-th transformer layer in $\epsilon_{\theta'_*}(\cdot)$ at the $t$-th timestep. $h_L, w_L, d_L$ are the height, width and channel of $\mathbf{E}^t$, respectively. $\epsilon_{\theta'_*}(\mathbf{z}_t|t)$ is the unconditional noise estimation, $\epsilon_{\theta'_*}(\mathbf{z}_t|\widehat{\mathbf{c}}, t)$ is the conditional noise estimation based on the textual embeddings $\widehat{\mathbf{c}}$, and $s = 7.5$ denotes the scale. Therefore, the noise estimation $\mathbf{E}^t_u \in \mathbb{R}^{h_L \times w_L \times d_L}$ for the $u$-th region condition $\{\widehat{\mathbf{p}}_u, \widehat{\mathbf{s}}_u\}$ can be formulated as follows:

$$\mathbf{E}^t_u = \epsilon_{\theta'_*}(\mathbf{z}_t|t) + s \cdot (\epsilon_{\theta'_*}(\mathbf{z}_t|[\widehat{\mathbf{c}}_u, \widehat{\mathbf{s}}_u], t) - \epsilon_{\theta'_*}(\mathbf{z}_t|t)), \tag{4}$$

where $\epsilon_{\theta'_*}(\mathbf{z}_t|[\widehat{\mathbf{c}}_u, \widehat{\mathbf{s}}_u], t)$ is the conditional noise estimation based on $[\widehat{\mathbf{c}}_u, \widehat{\mathbf{s}}_u]$. For multi-concept customization in the CIFC, we aggregate $U$ region noise estimations to address concept neglect:

$$\mathbf{E}^t_* = \alpha \mathbf{E}^t + \sum_{u=1}^U (1 - \alpha) \mathbf{E}^t_u \odot \widehat{\mathbf{m}}^L_u, \tag{5}$$

where $\widehat{\mathbf{m}}_u^L \in \mathbb{R}^{h_L \times w_L}$ is the binary region mask of the bounding box $\widehat{\mathbf{s}}_u$ in the $L$-th layer. $\alpha$ is the balance weight. Following [38, 33], we forward $\mathbf{E}_*^t$ to the denoising process for image synthesis.

## 5 Experiments

### 5.1 Experimental Setups

**Benchmark Dataset:** Motivated by [47, 11, 45], in this paper, we construct a new challenging concept-incremental learning (CIL) dataset including ten continuous text-guided concept customization tasks to illustrate the effectiveness of our model under the CIFC setting. In the CIL dataset, seven customization tasks have different object concepts (*i.e.*, V1 dog, V2 duck toy, V3 cat, V4 backpack, V5 teddy bear, V7 dog and V9 cat) from [40, 22], and the remaining three tasks have different style concepts (*i.e.*, V6, V8 and V10 styles) collected from website. Considering the practicality of the CIFC setting, we set about $3 \sim 5$ text-image pairs for each task. Particularly, we introduce some semantically similar concepts (*e.g.*, V1 and V7 dogs, V3 and V9 cats), making the CIL dataset more challenging under the CIFC setting.

**Implementation Details:** We utilize two popular diffusion models: Stable Diffusion (SD-1.5) [38] and SDXL [33] as the pretrained models to conduct comparison experiments. For fair comparisons, we train all SOTA comparison methods and our model using the same backbone and Adam optimizer, where the initial learning rate is $1.0 \times 10^{-3}$ to update textual embeddings, and $1.0 \times 10^{-4}$ to optimize the denoising UNet. For the low-rank matrices, we follow [11] to set $r = 4$. We empirically set $\gamma_1 = 0.1, \gamma_2 = 1.0$ in Eq. (2), $\alpha = 0.1$ in Eq. (5), and the training steps are 800.

**Evaluation Metrics:** After learning the final concept customization task under the CIFC setting, we conduct both the qualitative and quantitative evaluations on versatile generation tasks: single/multi-concept customization, custom image editing, and custom style transfer. For the quantitative evaluation, we follow [22] to use text-

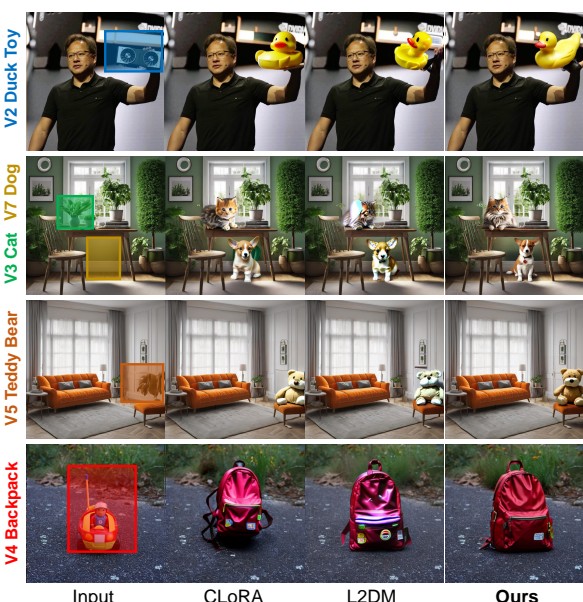

Figure 4: Comparisons of custom image editing.

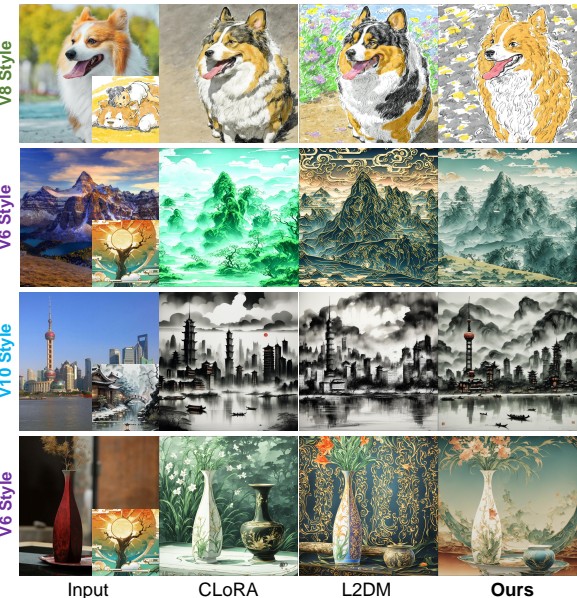

Figure 5: Comparisons of custom style transfer.

alignment (TA) and image-alignment (IA) as metrics. Specifically, for image-alignment (IA), we use the image encoder of CLIP [34] to evaluate the feature similarity between the synthesized image and original sample. For text-alignment (TA), we utilize the text encoder of CLIP [34] to compute the text-image similarity between the synthesized image and its corresponding prompt.

### 5.2 Qualitative Comparisons

To verify the superiority of our model under the CIFC setting, we introduce extensive qualitative comparisons, including single/multi-concept customization (see Figs. 2–3), custom image editing

Table 1: Comparisons (IA) of single-concept customization synthesized by SD-1.5 and SDXL.

| Methods | SD-1.5 [38] | | | | | | | | | | | SDXL [33] | | | | | | | | | | |
|---|---|---|---|---|---|---|---|---|---|---|---|---|---|---|---|---|---|---|---|---|---|---|
| | V1 | V2 | V3 | V4 | V5 | V6 | V7 | V8 | V9 | V10 | Avg. | V1 | V2 | V3 | V4 | V5 | V6 | V7 | V8 | V9 | V10 | Avg. |
| Finetuning | 77.6 | 82.2 | 79.0 | 77.6 | 79.6 | 62.9 | 71.5 | 53.7 | 81.4 | 72.1 | 73.7 | 62.0 | 70.8 | 79.1 | 73.4 | 76.4 | 67.5 | 76.8 | 57.4 | 77.1 | 74.8 | 71.5 |
| EWC [20] | 78.7 | 83.8 | 80.4 | 80.3 | 80.7 | 64.0 | 76.5 | 57.1 | 84.4 | 73.1 | 75.9 | 83.6 | 80.5 | 84.6 | 80.8 | 79.2 | 70.1 | 80.5 | 61.2 | 79.5 | 75.8 | 77.6 |
| LWF [25] | 80.4 | 79.7 | 80.9 | 77.4 | 80.9 | 61.8 | 73.2 | 53.5 | 78.1 | 74.7 | 74.1 | 84.0 | 81.2 | 84.2 | 81.7 | 79.7 | 68.1 | 77.1 | 60.1 | 76.3 | 72.7 | 76.5 |
| LoRA-M [69] | 80.0 | 84.2 | 79.1 | 76.5 | 82.7 | 65.7 | 70.1 | 54.7 | 79.5 | 74.1 | 74.6 | 82.6 | 79.9 | 84.5 | 80.1 | 80.9 | 57.8 | 77.0 | 54.0 | 71.8 | 74.0 | 74.3 |
| LoRA-C [69] | 80.1 | 84.1 | 79.8 | 76.6 | 82.9 | 65.9 | 70.8 | 54.9 | 79.9 | 74.4 | 74.9 | 82.8 | 80.4 | 84.8 | 80.0 | 81.0 | 58.2 | 76.8 | 54.5 | 72.2 | 73.9 | 74.5 |
| CLoRA [45] | 83.2 | 83.4 | 81.1 | 80.6 | 84.9 | 66.3 | 76.2 | 58.1 | 83.0 | 72.1 | 76.9 | 83.4 | 81.3 | 85.8 | 80.1 | 79.0 | 70.4 | 81.2 | 61.7 | 78.5 | 76.7 | 77.8 |
| L2DM [47] | 78.7 | 86.3 | 76.6 | 80.7 | 86.8 | 70.8 | 70.0 | 59.3 | 77.7 | 74.1 | 76.1 | 84.6 | 79.5 | 81.9 | 75.5 | 82.1 | 69.2 | 80.9 | 63.8 | 77.0 | 76.4 | 77.1 |
| **CIDM (Ours)** | **83.6** | **86.4** | **82.9** | **80.8** | 86.5 | 69.5 | 73.7 | 56.9 | 82.4 | **75.9** | **78.0** | **87.1** | 82.1 | 88.5 | 84.9 | 85.8 | 68.3 | **82.0** | 62.4 | 76.9 | 76.6 | **79.5** |

Table 2: Comparisons (TA) of single-concept customization synthesized by SD-1.5 and SDXL.

| Methods | SD-1.5 [38] | | | | | | | | | | | SDXL [33] | | | | | | | | | | |
|---|---|---|---|---|---|---|---|---|---|---|---|---|---|---|---|---|---|---|---|---|---|---|
| | V1 | V2 | V3 | V4 | V5 | V6 | V7 | V8 | V9 | V10 | Avg. | V1 | V2 | V3 | V4 | V5 | V6 | V7 | V8 | V9 | V10 | Avg. |
| Finetuning | 64.4 | 74.6 | 69.4 | 68.6 | 75.0 | 70.0 | **76.7** | 69.2 | 65.4 | 67.2 | 70.0 | 54.8 | 77.5 | 72.2 | 85.0 | 80.5 | 76.2 | **79.7** | 73.6 | 77.6 | 76.3 | 75.3 |
| EWC [20] | 67.1 | 77.5 | 72.7 | 77.9 | 76.7 | 72.3 | 74.2 | 72.0 | 66.0 | 70.4 | 72.7 | 71.4 | 79.8 | 72.8 | 84.4 | 79.5 | 73.9 | 76.7 | 77.0 | 78.3 | 77.6 | 77.1 |
| LWF [25] | 70.8 | 75.2 | 71.0 | 77.4 | 76.0 | 71.7 | 76.3 | 72.9 | 72.5 | 70.0 | 73.4 | 75.8 | 76.9 | 76.0 | 83.6 | 82.9 | 75.1 | 76.7 | 74.3 | 79.1 | 76.8 | 77.7 |
| RPY [25] | 68.1 | 76.2 | 70.1 | 78.4 | 75.7 | 69.3 | 74.8 | 70.5 | 65.8 | 68.6 | 71.8 | 69.3 | 81.0 | 71.9 | 87.3 | 78.8 | 71.5 | 76.4 | 75.9 | 79.7 | 76.2 | 76.8 |
| CLoRA [45] | 69.4 | 78.0 | 74.1 | 78.8 | 76.4 | 69.6 | 76.7 | 73.9 | 69.0 | 71.8 | 73.6 | 71.8 | 80.1 | 71.1 | 87.7 | 81.2 | 74.6 | 77.8 | 77.7 | 80.1 | 75.9 | 77.8 |
| L2DM [47] | 68.6 | 79.5 | 70.1 | 73.0 | 76.7 | 67.7 | 75.9 | 74.1 | 71.8 | 69.4 | 72.7 | 72.6 | 78.4 | 78.5 | 85.0 | 81.5 | 73.5 | 78.6 | 79.1 | 81.9 | 77.8 | 78.7 |
| **CIDM (Ours)** | **75.3** | 78.1 | 74.0 | 81.1 | 78.2 | 70.1 | 74.7 | **74.3** | 73.5 | 70.2 | **74.8** | 74.9 | 79.6 | 74.5 | 86.7 | 83.5 | 79.8 | 78.2 | 83.1 | 81.4 | 78.5 | **80.0** |

(see Fig. 4), and custom style transfer (see Fig. 5). 1) As presented in Fig. 2, the proposed model achieves the best performance for single-concept customization by addressing catastrophic forgetting of old personalized concepts and preserving superior attributes of each learned concept. 2) For multi-concept customization, we incorporate the regionally controllable sampling module proposed in [11] into existing comparison methods to fairly compare them with our model. As shown in Fig. 3, all comparison methods significantly suffer from the challenge of concept neglect, and they are difficult to generate multiple object concepts. In contrast, our proposed model can effectively tackle the challenge of concept neglect via the proposed context-controllable synthesis strategy. 3) To perform custom image editing, we introduce Anydoor [3] as a plug-in for all comparison methods in Fig. 4. The qualitative comparisons in Fig. 4 reflect the effectiveness of our model in custom image editing. Such substantial improvement benefits from the superior performance of our concept consolidation loss to preserve unique identity of each learned concept under the CIFC setting. 4) Fig. 5 shows qualitative comparisons of custom style transfer. Our proposed model presents the best performance for image-to-image style transfer, since the elastic weight aggregation can preserve style integrity in the CIFC setting. The above qualitative comparisons in Figs. 2–5 verify that the proposed model can significantly surpass existing latent diffusion models to tackle the CIFC problem.

## 5.3 Quantitative Comparisons

To analyze quantitative comparisons between our model and SOTA methods, we follow [11, 22, 45, 47] to introduce 20 evaluation prompts for each concept and generate 50 images for each evaluation prompt, resulting in a total of 1,000 synthesized images. Then quantitative evaluation is con-

Table 3: Ablation studies of single-concept customization.

| Variants | TSP | TSH | EWA | V1–V5 | V6–V10 | Avg. |
|---|---|---|---|---|---|---|
| Baseline | | | | 80.4 | 68.8 | 74.6 |
| Baseline w/ EWA | | | ✓ | 83.7 | 70.8 | 77.3 |
| Ours w/o TSP | | ✓ | ✓ | 83.7 | 71.0 | 77.4 |
| Ours w/o TSH | ✓ | | ✓ | 83.9 | 71.4 | 77.7 |
| **CIDM (Ours)** | ✓ | ✓ | ✓ | **84.0** | **71.7** | **77.9** |

ducted on these 1,000 images. As shown in Tabs. 1–2, we can observe that our CIDM outperforms all comparison methods by $1.1\% \sim 8.0\%$ in terms of image-alignment (IA) and $1.2\% \sim 4.8\%$ in terms of text-alignment (TA). It illustrates that our model can effectively preserve unique identity of each learned concept to tackle the CIFC problem. Compared to other SOTA methods, our proposed model achieves better performance in mitigating catastrophic forgetting, as the introduced concept consolidation loss captures task-specific information and task-shared knowledge through two-step

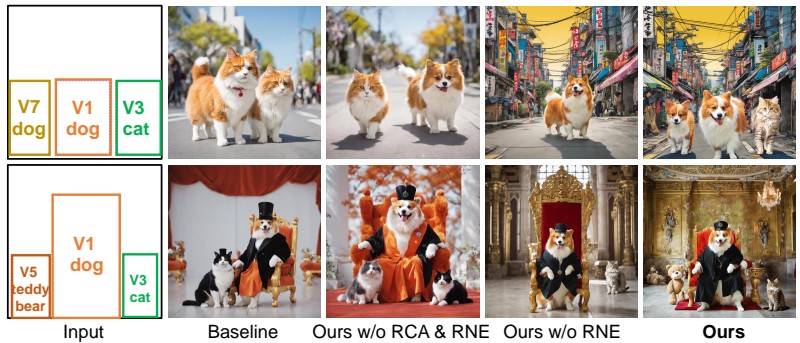

Figure 6: Ablation analysis of CSS in multi-concept customization.

optimization during training. Moreover, the elastic weight aggregation module proposed in this paper effectively merges previous personalized concepts for customization during inference.

## 5.4 Ablation Studies

This subsection analyzes the effectiveness of each module in our model: elastic weight aggregation (EWA), context-controllable synthesis (CCS), task-specific knowledge (TSP) and task-shared knowledge (TSH) in the concept consolidation loss (CCL). Tab. 3 presents the ablation studies of single-concept customization in terms of IA. When compared with Baseline, the performance of our model improves by $0.2\% \sim 3.3\%$ in terms of IA, after we add the proposed TSP, TSH and EWA modules. It demonstrates the effectiveness of our model in resolving the CIFC problem by addressing the catastrophic forgetting and concept neglect. As shown in Fig. 6, we analyze the effectiveness of CCS in multi-concept customization, where CCS

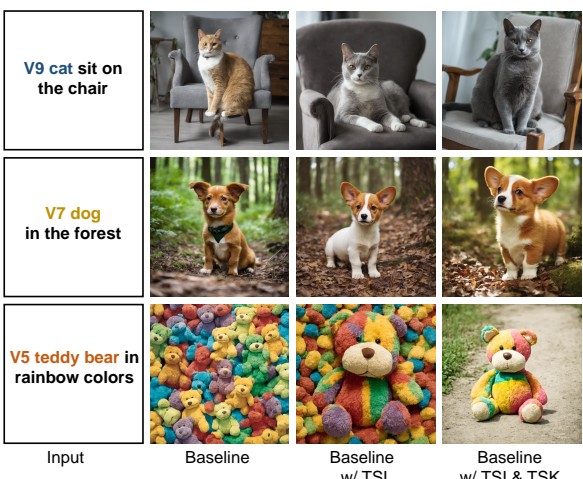

Figure 7: Ablation studies of the TSP and TSH.

includes the layer-wise regional cross-attention (RCA) and the region noise estimation (RNE) modules. In Fig. 6, the performance of our model decreases substantially when removing the RCA and RNE modules. It verifies the effectiveness of our CCS startegy in addressing conditional generation and concept neglect. Moreover, Fig. 7 shows ablation analysis about TSP and TSH. It illustrates that our model can capture task-specific information within each customization task and explore task-shared knowledge across different tasks to tackle the CIFC problem via optimizing Eq. (2).

## 6 Conclusion

In this paper, we propose a novel Concept-Incremental text-to-image Diffusion Model (CIDM) to address a practical Concept-Incremental Flexible Customization (CIFC) problem, where the CIFC problem has two major challenges: catastrophic forgetting and concept neglect. Specifically, we devise a concept consolidation loss and an elastic weight aggregation module to respectively resolve catastrophic forgetting during training and inference. They can capture task-specific/task-shared knowledge and aggregate all low-rank weights of old concepts according to their contributions in the CIFC. To address concept neglect, we propose a new context-controllable synthesis strategy, which can control the contexts of synthesized images according to users-provided conditions. Extensive experiments on versatile customization tasks (single/multi-concept customization, custom image editing and style transfer) show the superior performance of our CIDM in tackling the CIFC problem compared to SOTA methods. In the future, we will leverage multi-modal large language models to address the CIFC problem and apply the proposed model to personalized video generation.

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

# A Appendix

## A.1 Optimization pipeline

As shown in **Algorithm** 1, we introduce the detailed algorithm pipeline of our proposed CIDM to tackle the CIFC problem. For the first text-guided concept customization task ($g = 1$) in the training process, we only utilize Eq. (1) to optimize the textual embeddings of layer-wise concept tokens and low-rank weight $\triangle\theta_g = \{\triangle\mathbf{W}_g^l\}_{l=1}^L$. When $g \geq 2$, we devise a two-step optimization strategy in each training batch to efficiently train our CIDM via Eq. (2). It can continually learn new personalized concepts under the CIFC setting, while overcoming catastrophic forgetting of old personalized concepts by exploring both task-specific and task-shared knowledge during training.

1) In the first step, we fix $\mathbf{H}_i^l$ and $\mathbf{W}_*^l$ to update learnable layer-wise concept tokens and low-rank weight $\triangle\theta_g = \{\triangle\mathbf{W}_g^l\}_{l=1}^L$ via minimizing the loss $\mathcal{L}_{\text{CCL}}$ in Eq. (2). Particularly, the loss $\mathcal{L}_{\text{CCL}}$ can capture task-specific knowledge (*i.e.* unique concept attributes within each customization task).

2) In the second step, in order to explore task-shared knowledge across different tasks, we fix the learnable layer-wise concept tokens and low-rank weight $\triangle\theta_g$ to update $\mathbf{H}_i^l$ and $\mathbf{W}_*^l$ via minimizing the regulaizer $\mathcal{R}_2 = \sum_{i=1}^g \sum_{l=1}^L \|\triangle\mathbf{W}_i^l - \mathbf{H}_i^l\mathbf{W}_*^l\|_F^2$. As a result, the gradients of $\mathcal{R}_2$ with respect to $\mathbf{H}_i^l$ and $\mathbf{W}_*^l$ are formulated as follows:

$$\frac{\partial\mathcal{R}_2}{\partial\mathbf{H}_i^l} = -2(\triangle\mathbf{W}_i^l - \mathbf{H}_i^l\mathbf{W}_*^l)(\mathbf{W}_*^l)^\top, \quad \frac{\partial\mathcal{R}_2}{\partial\mathbf{W}_*^l} = -2(\mathbf{H}_i^l)^\top(\triangle\mathbf{W}_i^l - \mathbf{H}_i^l\mathbf{W}_*^l). \tag{6}$$

Given the gradients $\partial\mathcal{R}_2/\partial\mathbf{H}_i^l$ and $\partial\mathcal{R}_2/\partial\mathbf{W}_*^l$ in Eq. (6), we can iteratively update $\mathbf{H}_i^l$ and $\mathbf{W}_*^l$ via the following objective:

$$\mathbf{H}_i^l := \mathbf{H}_i^l + \eta\frac{\partial\mathcal{R}_2}{\partial\mathbf{H}_i^l}, \quad \mathbf{W}_*^l := \mathbf{W}_*^l + \eta\frac{\partial\mathcal{R}_2}{\partial\mathbf{W}_*^l}, \tag{7}$$

where $\eta$ denotes the learning rate to update $\mathbf{H}_i^l$ and $\mathbf{W}_*^l$. In this paper, we set the value of $\eta$ to be the same as the learning rate of the Adam optimizer.

## A.2 Experiment Setups

This subsection introduces more details about dataset, comparison methods and evaluation metrics.

**Dataset:** Inspired by [47, 11, 45], we construct a new challenging concept-incremental learning (CIL) dataset including ten continuous text-guided concept customization tasks to verify the effectiveness of our model under the CIFC setting. In the CIL dataset, as shown in Fig. 8, seven tasks have different object concepts (*i.e.*, V1 dog, V2 duck toy, V3 cat, V4 backpack, V5 teddy bear, V7 dog and V9 cat) from [40, 22], and the remaining three tasks have different style concepts (*i.e.*, V6, V8 and V10 styles) collected from website. Considering the practicality of the CIFC setting, we set about $3 \sim 5$ text-image pairs for each task, where the text prompts are extracted by BLIP [23]. Particularly, different from L2DM [47] and Mix-of-Show [11], we introduce some semantically similar concepts (*e.g.*, V1 and V7 dogs, V3 and V9 cats), making the CIL dataset more challenging in the CIFC setting.

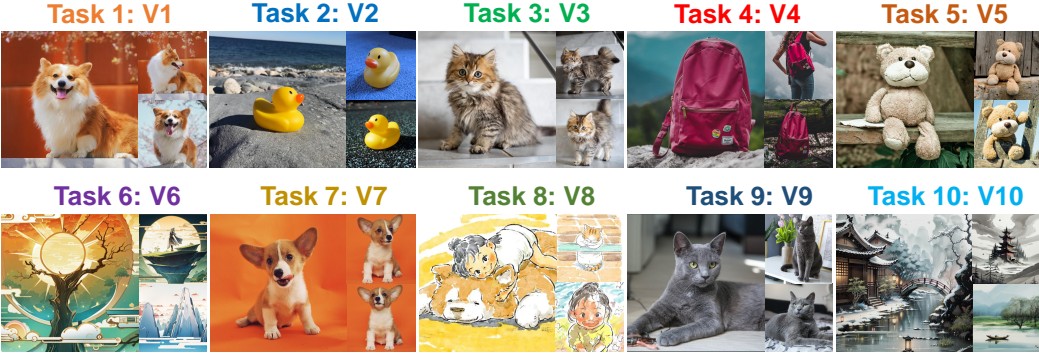

Figure 8: Visualization of some examples in ten consecutive concept customization tasks.

| Prompts for Pets | Prompts for Objects | Prompts for Styles |
|---|---|---|
| 1. A [V*] in the swimming pool | 1. A [V*] in the ocean | 1. A [V*] of a tree, mountain and moon |
| 2. A [V*] in front of Eiffel tower | 2. A [V*] near the Eiffel tower | 2. A [V*] of a wooden house |
| 3. A [V*] near the mount fuji | 3. A [V*] near the mount fuji | 3. A [V*] of a bridge over the river with mountain background |
| 4. A [V*] in the forest | 4. A [V*] in the forest | 4. A [V*] of a waterfalls appearing among mountains |
| 5. A [V*] walking on the street | 5. A [V*] in times square | |
| 6. A [V*] cyberpunk 2077, 4K, 3d render in unreal engine | 6. A [V*] cyberpunk 2077, 4K, 3d render in unreal engine | 5. A [V*] of sun rises among mountains |
| 7. A watercolor painting of a [V*] | 7. A watercolor painting of a [V*] | 6. A [V*] of flowers, grass and river |
| 8. A painting of a [V*] in the style of Vincent Van Gogh | 8. A painting of a [V*] in the style of Vincent Van Gogh | 7. A [V*] of a boat on a lake |
| 9. A painting of a [V*] in the style of Claude Monet | 9. A painting of a [V*] in the style of Claude Monet | 8. A [V*] of a dragon with the cloud background |
| 10. A [V*] in the style of Pixel Art | 10. A [V*] in the style of Pixel Art | 9. A [V*] of a cliff with a sky and moon background |
| 11. A [V*] sit on the chair | 11. A [V*] on a table | 10. A [V*] of mountains, moon and clouds |
| 12. A [V*] on the boat | 12. A [V*] on a chair | 11. A [V*] of a smile girl |
| 13. A [V*] wearing a headphone | 13. A [V*] on a skateboard | 12. A [V*] of a dog |
| 14. A [V*] wearing a sunglass | 14. A child is playing a [V*] | 13. A [V*] > of a cat |
| 15. A [V*] playing with a ball | 15. A [V*] on a carpet | 14. A [V*] of a snow, tree and clouds |
| 16. A sad [V*] | 17. A close view of [V*] | 15. A [V*] of a panda |
| 17. An angry [V*] | 16. A top view of [V*] | 16. A [V*] in the style of Vincent Van Gogh |
| 18. A running [V*] | 18. A [V*] in rainbow colors | 17. A [V*] in the style of Claude Monet |
| 19. A jumping [V*] | 19. A fallen [V*] | 18. A watercolor painting of a [V*] |
| 20. A [V*] is lying down | 20. A broken [V*] | 19. A [V*] in the style of Pixel Art |
| | | 20. A [V*] cyberpunk 2077, 4K, |

Figure 9: Descriptions of text prompts used in this paper.

---

**Algorithm 1:** Algorithm Pipeline of The Proposed CIDM.

---

**Initialize:** The pretrained denoising UNet $\epsilon_\theta(\cdot)$, a sequence of text-guided concept customization tasks $\mathcal{T} = \{\mathcal{T}_g\}_{g=1}^G$ and the pretrained CLIP text encoder $\Gamma(\cdot)$;
▷ **Training for The $g$-th Task:**
**Initialize:** Low-rank weight $\triangle\theta_g = \{\triangle\mathbf{W}_g^l\}_{l=1}^L$;
**for** $(\mathbf{x}_g^k, \mathbf{p}_g^k, \mathbf{y}_g^k)$ in $\mathcal{T}_g$ **do**
    Initialize the layer-wise text prompts $\{\mathbf{p}_g^{k,l}\}_{l=1}^L$;
    Obtain the layer-wise textual embeddings $\mathbf{c}_g^{k,l}$ via $\Gamma(\mathbf{p}_g^{k,l})$;
    **if** $g \geq 2$ **then**
        Update the learnable layer-wise concept tokens and $\triangle\theta_g$ via Eq. (2) when fixing $\mathbf{H}_i^l$ and $\mathbf{W}_*^l$;
        Update $\mathbf{H}_i^l$ and $\mathbf{W}_*^l$ via Eq. (7) when fixing the learnable layer-wise concept tokens and $\triangle\theta_g$;
    **else**
        Update the learnable layer-wise concept tokens and $\triangle\theta_g$ via Eq. (1);
**Return:** Task learner $Z_g = \{\triangle\mathbf{W}_g^l, \widehat{\mathbf{y}}_g^l\}_{l=1}^L$ of the $g$-th concept customization task $\mathcal{T}_g$.
▷ **Inference:**
**Initialize:** All learned task learners $\{\mathcal{Z}_i\}_{i=1}^G$; initial text prompt $\widehat{\mathbf{p}}$ and region conditions $\{\widehat{\mathbf{p}}_u, \widehat{\mathbf{s}}_u\}_{u=1}^U$;
**for** $l$ in $\{1, 2 \ldots, L\}$ **do**
    Update the low-rank weights $\triangle\widehat{\mathbf{W}}^l$ via Eq. (3) to obtain a new denoising UNet $\epsilon_{\theta_*'}(\cdot)$;
**for** $t$ in $\{T, T-1, \cdots, 1\}$ **do**
    **for** $\{\widehat{\mathbf{p}}_u, \widehat{\mathbf{s}}_u\}$ in $\{\widehat{\mathbf{p}}_u, \widehat{\mathbf{s}}_u\}_{u=1}^U$ **do**
        **for** $l$ in $\{1, 2 \ldots, L\}$ **do**
            Obtain the $u$-th region feature $\mathbf{f}_u^l$; Replace the values of $\mathbf{f}^l$ inside the bounding box $\widehat{\mathbf{s}}_u$ with $\mathbf{f}_u^l$;
        Obtain the $u$-th region noise estimation $\mathbf{E}_u^t$ via Eq. (4);
    Use $\mathbf{E}_*^t$ in Eq. (5) to update the noisy latent feature from $\mathbf{z}_t$ to $\mathbf{z}_{t-1}$;
**Return:** The generated image $\widehat{\mathbf{x}} = \mathcal{D}(\mathbf{z}_1)$.

---

**Comparison Methods:** To comprehensively demonstrate the superior performance of our model, we introduce eight SOTA comparison methods, including four continual learning-based methods (*i.e.*, Finetuning, EWC [20], LWF [25] and RPY [25]), two continual diffusion models (*i.e.*, CLoRA [45] and L2DM [47]), and two multi-LoRA composition methods (LoRA-M [69] and LoRA-C [69]). Specifically, Finetuning aims to optimize LoRA layers to learn multiple personalized concepts in a concept-incremental manner. To address catastrophic forgetting, EWC [20] applies an elastic regularizer to penalize the network parameters. Besides, LWF [25] stores the old training data to perform knowledge distillation on the current task, while RPY [25] replays the old concepts [20] to

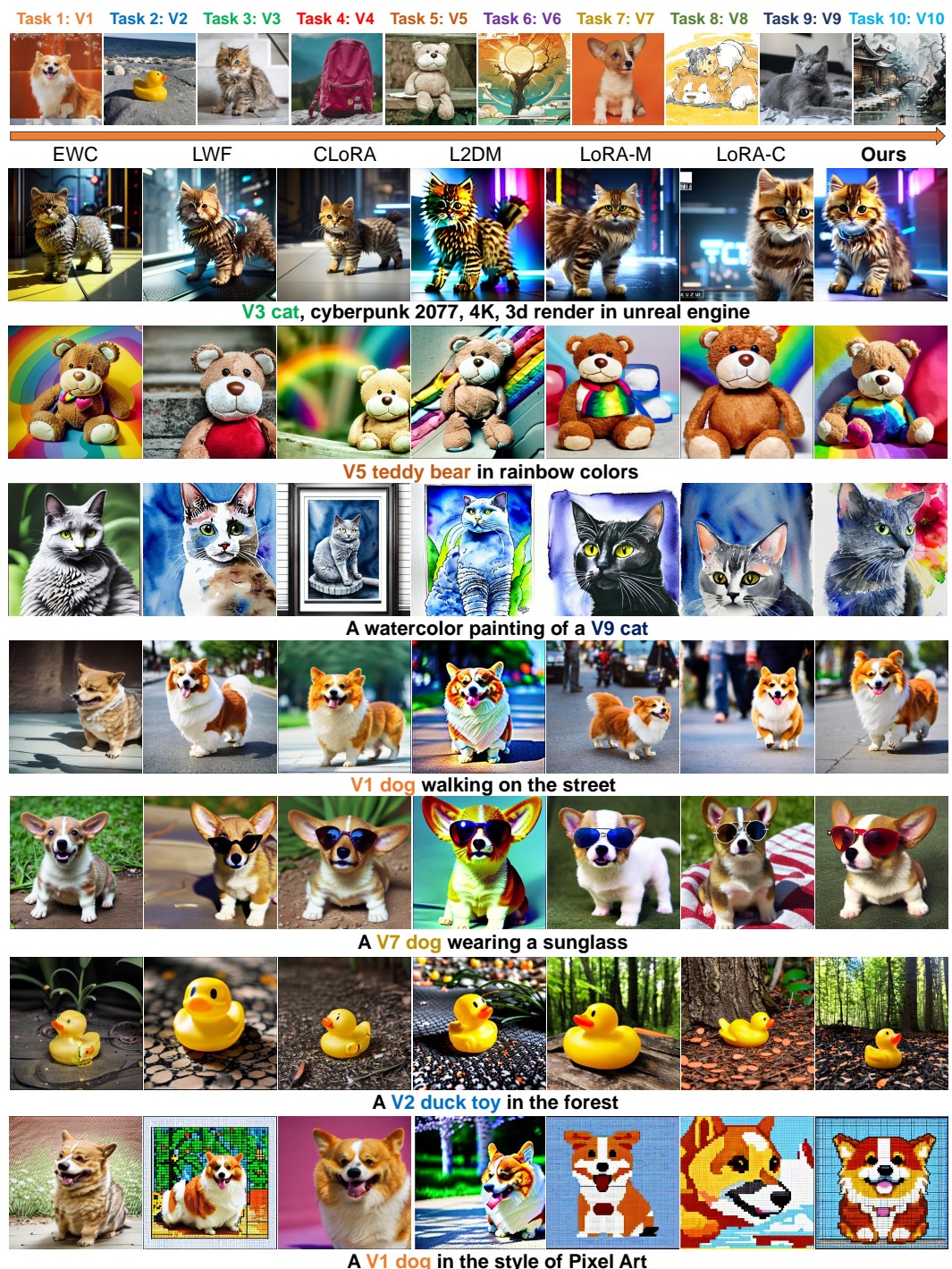

Figure 10: Some qualitative comparisons of single-concept customization generated by SD-1.5 [38].

tackle the CIFC setting. CLoRA [45] proposes a self-regularized low-rank adaption to continually learn new personalized concepts. L2DM [47] builds a long-term memory bank to reconstruct old personalized concepts, and performs knowledge distillation to mitigate catastrophic forgetting. LoRA-M [69] equally amalgamates all LoRA layers to retrain the latent diffusion model. LoRA-C [69] focuses on exploring contributions of different LoRA layers for multi-concept composition.

**Evaluation Metrics:** After learning the final concept customization task under the CIFC setting, we conduct both the qualitative and quantitative evaluations on versatile generation tasks: single/multi-concept customization, custom image editing, and custom style transfer. For the quantitative evalua-

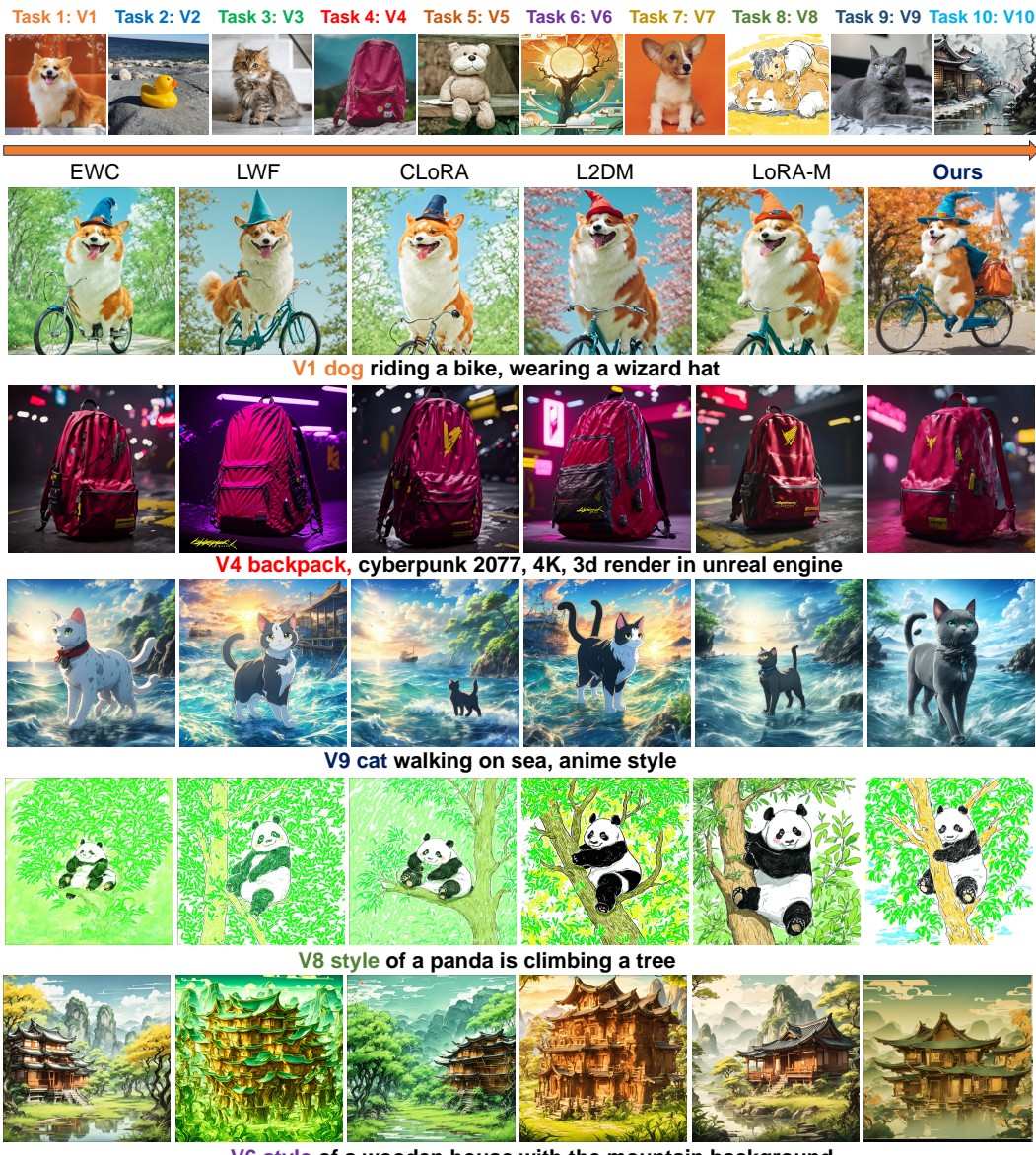

Figure 11: Some qualitative comparisons of single-concept customization generated by SDXL [33].

tion, we follow [22] to use text-alignment (TA) and image-alignment (IA) as metrics. Specifically, for image-alignment (IA), we use the image encoder of CLIP [34] to evaluate the feature similarity between the synthesized image and original sample. For text-alignment (TA), we utilize the text encoder of CLIP [34] to compute the text-image similarity between the synthesized image and its corresponding prompt. As shown in Fig. 9, we roughly categorize prompts into three categories according to the learned concepts. To analyze quantitative comparisons between our model and SOTA methods, inspired by [11, 22, 45, 47], we input an evaluation prompt and a unify negative prompt to sample 50 images. Therefore, given 20 evaluation prompts in this paper, we can generate 1,000 images for each concept to evaluate performance in terms of TA and IA.

## A.3   Implementation Details

In this paper, we use two popular diffusion models: Stable Diffusion (SD-1.5) [38] and SDXL [33] as the pretrained models to conduct comparison experiments. For fair comparisons, we train all SOTA comparison methods and our model using the same diffusion model and Adam optimizer, where

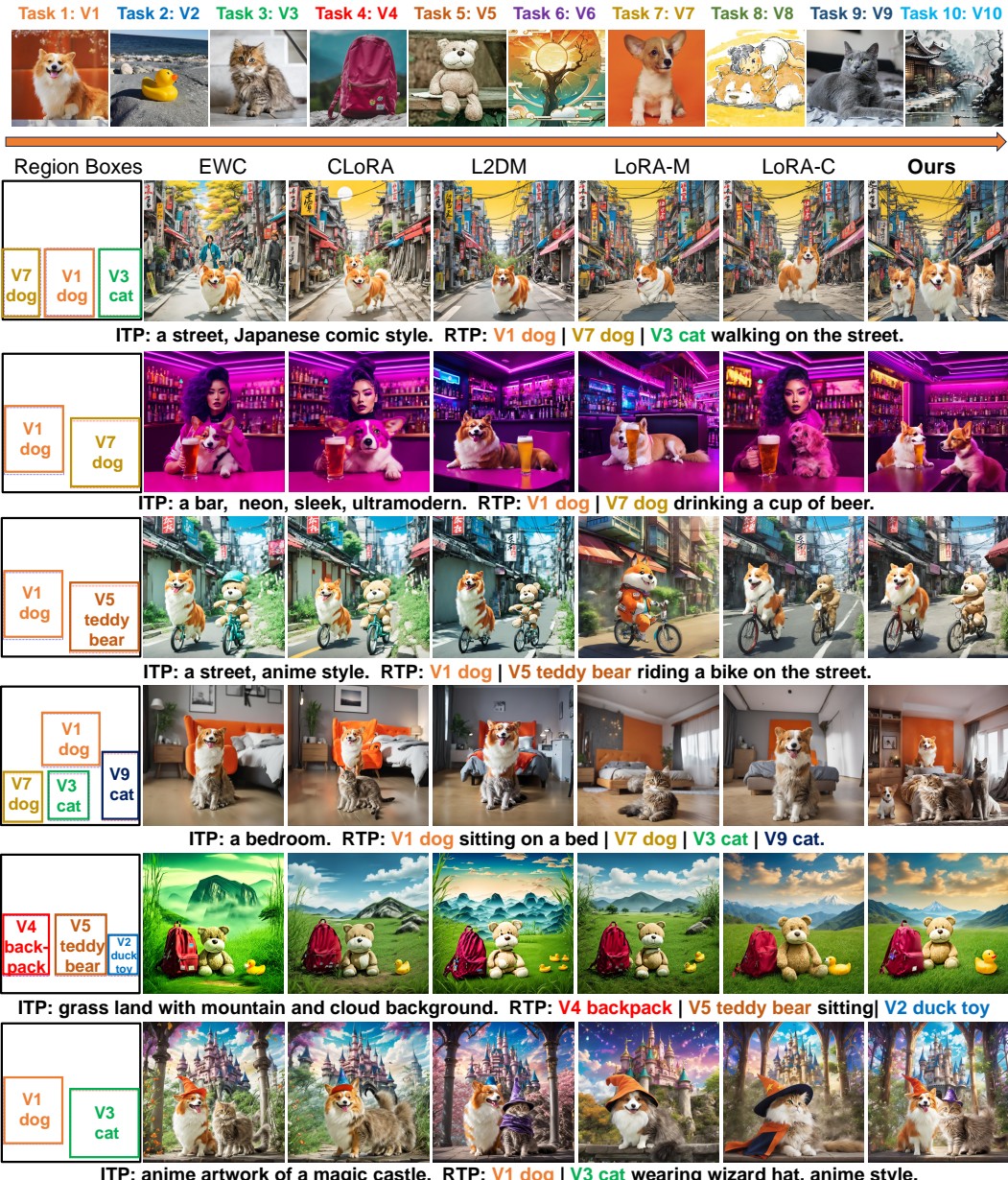

Figure 12: Some qualitative comparisons of multi-concept customization generated by SDXL [33], where ITP indicates the initial text prompt, and RTP denotes the region text prompt.

the initial learning rate is $1.0 \times 10^{-3}$ to update textual embeddings, and $1.0 \times 10^{-4}$ to optimize the denoising UNet. For the low-rank matrices, we follow [11] to set $r = 4$. Moreover, we empirically set $\gamma_1 = 0.1, \gamma_2 = 1.0$ in Eq. (2), $\alpha = 0.1$ in Eq. (5), and the training steps are 800. In this paper, we train our model on two NVIDIA RTX 4090 GPUs. In each text-guided concept customization task, we finetune both the textual embeddings of layer-wise concept tokens and low-rank weights via the proposed two-step optimization strategy in Sec. A.1. For example, given a textual description $\mathbf{p}_g^k$ (photo of a $[V_*] [V_{\text{dog}}]$) of image $\mathbf{x}_g^k$ in the $g$-th task, its text prompt of the $l$-th layer is defined as $\mathbf{p}_g^{k,l}$ (photo of a $[V_*^l] [V_{\text{dog}}^l]$). After using $[V_*] [V_{\text{dog}}]$ to initialize $[V_*^l] [V_{\text{dog}}^l]$ in the $l$-th transformer layer, we only update the layer-wise concept tokens (e.g., $[V_*^l] [V_{\text{dog}}^l]$). For fair comparisons in the CIFC setting, all comparison methods and our model use the same prompt augmentation and image augmentation during the training phase. Additionally, we set the random seed as 0 for all comparison experiments. More importantly, we have no prior knowledge about task order, task quantity and data distributions in the CIFC problem.

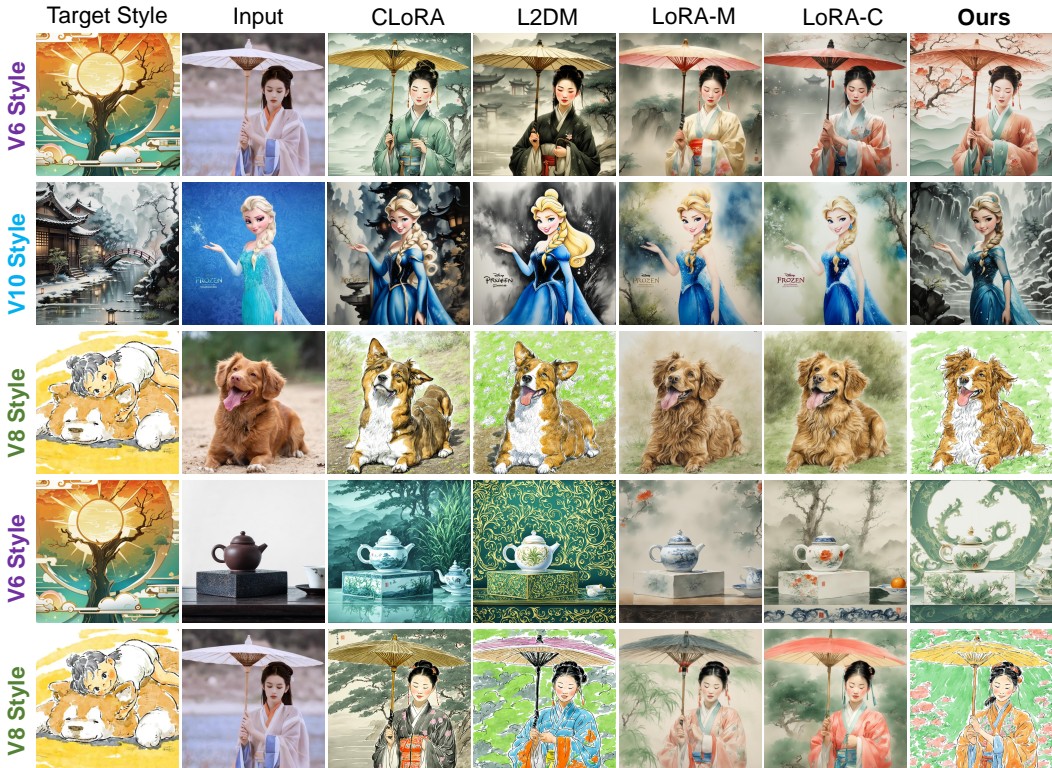

Figure 13: Some qualitative comparisons of custom style transfer under the CIFC setting.

## A.4 More Qualitative Comparisons

**Single-Concept Customization:** As presented in Figs. 10–11, we conduct extensive qualitative comparisons of single-concept customization, when using Stable-Diffusion (SD-1.5) [38] and SDXL [33] as the pretrained denoising UNet. In Fig. 11, given a text prompt (*e.g.*, "V1 dog riding a bike, wearing a wizard hat") for inference, we observe that our model can effectively follow the prompt instruction to synthesize images that preserve the superior identity of the personalized concept "V1 dog". In contrast, existing comparison methods suffer from significant catastrophic forgetting, generating unpleasant images with distorted objects and unmatched identity.

**Multi-Concept Customization:** Fig. 12 presents comprehensive comparison experiments of multi-concept customization. For fair comparisons with existing methods, we apply the region-aware cross-attention module proposed in Mix-of-Show [11] into all comparison methods. The qualitative comparisons in Fig. 12 demonstrate the superior performance of our model in multi-concept customization, when continually learning new personalized concepts under the CIFC setting. Particularly, given an initial text prompt (ITP) and some region text prompts (RTP) with users-provided bounding boxes, all comparison methods exhibit significant concept neglect and they are difficult to perform multi-concept customization. On the one hand, it is evident that all comparison methods fail to preserve the identity of learned concepts due to catastrophic forgetting. On the other hand, these methods neglect some important object concepts during the synthesis process. For example, users may want to generate an image of a V1 dog and a V7 dog drinking a cup of beer, but none of the comparison methods can generate the V1 and V7 dogs, as shown in Fig. 12.

**Custom Style Transfer:** As shown in Fig. 13, we introduce some comparisons of custom style transfer to evaluate the effectiveness of our proposed CIDM in addressing the CIFC problem. From Fig. 13, we conclude that our model achieves the best generation performance according to user-provided style concepts, compared to other existing methods. It verifies that the designed elastic weight aggregation module plays a crucial role in preserving distinctive style attributes in the CIFC setting. In contrast, existing comparison methods struggle to explore the identity of different style concepts, due to the catastrophic forgetting and concept neglect in the CIFC setting.

Table 4: Ablation studies (IA) of single-concept customization generated by SD-1.5 [38].

| Variants | TSP | TSH | EWA | V1 | V2 | V3 | V4 | V5 | V6 | V7 | V8 | V9 | V10 | Avg. |
|---|---|---|---|---|---|---|---|---|---|---|---|---|---|---|
| Baseline | | | | 80.0 | 84.2 | 79.1 | 76.5 | 82.7 | 65.7 | 70.1 | 54.7 | 79.5 | 74.1 | 74.6 |
| Baseline w EWA | | | ✓ | 82.7 | 85.8 | 81.9 | 81.4 | 86.5 | 68.2 | 72.8 | 56.1 | 81.8 | 75.2 | 77.3 |
| Ours w/o TSP | | ✓ | ✓ | 82.9 | 85.9 | 81.8 | **81.5** | 86.3 | 68.5 | 72.6 | 56.7 | 82.2 | 75.2 | 77.4 |
| Ours w/o TSH | ✓ | | ✓ | 83.3 | 86.3 | **82.9** | 80.6 | **86.6** | 69.4 | 73.2 | 56.7 | 82.3 | 75.6 | 77.7 |
| **CIDM (Ours)** | ✓ | ✓ | ✓ | **83.6** | **86.4** | **82.9** | 80.8 | 86.5 | **69.6** | **73.7** | **57.0** | **82.5** | **75.9** | **77.9** |

## A.5 More Ablation Studies

This subsection analyzes the effectiveness of each module in our model: elastic weight aggregation (EWA), task-specific knowledge (TSP) and task-shared knowledge (TSH) in the concept consolidation loss (CCL). As shown in Tab. 4, we introduce some quantitative ablation studies (IA) of the single-concept customization generated by SD-1.5 [38]. As can be seen from Tab. 4, our model significantly outperforms other variants by $0.2\% \sim 3.3\%$ in terms of image-alignment (IA). These ablation experiments validate that our model can effectively explore task-shared knowledge across different customization tasks to tackle the CIFC problem during training. Moreover, it also verifies that the proposed elastic weight aggregation module is effective to address catastrophic forgetting during inference by aggregating different low-rank weighs according to their contributions.

## A.6 Societal Impact and Limitations

**Societal Impact:** To tackle the practical concept-incremental flexible customization (CIFC) problem, our proposed concept-incremental text-to-image diffusion model (CIDM) aims to continually learn new personalized concepts in a concept-incremental manner for versatile customization (*e.g.*, single/multi-concept customization, custom image editing and custom style transfer), while tackling catastrophic forgetting and concept neglect on old concepts. In particular, it allows users to continually generate a sequence of images using their new personalized concepts. Additionally, users can control the contexts of synthesized images according to their own preferences.

Generally, the proposed CIFC setting can enable the creation of highly personalized content for various applications such as marketing, entertainment, and education. This can lead to more engaging and relevant experiences for users. More importantly, artists, designers, and content creators can benefit from tools that adapt to their unique styles and preferences over time. This can foster innovation and creativity by providing customized suggestions and automating repetitive tasks. Additionally, it can be tailored to meet the needs of different user groups, including those with disabilities. For example, generating personalized educational materials can cater to diverse learning styles and needs. Consequently, the CIFC problem proposed In this paper is worth being studied. More importantly, our proposed CIDM can achieve state-of-the-art performance in addressing the CIFC problem, highlighting the importance of this work in promoting the development of text-to-image diffusion models. In the CIFC, the use of personalized data to train our model may raise privacy issues. However, this is a common concern for all latent diffusion models (LDMs). Ensuring that user data is handled securely and ethically is paramount to prevent misuse or unauthorized access, safeguarding both privacy and trust.

**Limitations:** Although we construct a concept-incremental dataset containing four semantically similar concepts to verify the effectiveness of our model, it will still suffer from catastrophic forgetting and concept neglect as the number of semantically similar concepts increases. The main limitation of this work is that our model struggle to continually learn large-scale semantically similar concepts provided by users. Thus, we will explore how to increase the scalability of our model in the future.

