# OpenReview forum: "How to Continually Adapt Text-to-Image Diffusion Models for Flexible Customization?"
_NeurIPS.cc/2024/Conference — NeurIPS 2024 poster_

### Official Review · Reviewer_SpKd · 2024-07-09

**Soundness:** 3
**Presentation:** 2
**Contribution:** 3
**Rating:** 5
**Confidence:** 4

**Summary:**

This paper introduces a novel approach called CIDM to address the challenge of Concept-Incremental Flexible Customization in text-to-image generation. The authors identify two key issues in CIFC: catastrophic forgetting of previously learned concepts and concept neglect during multi-concept composition. The proposed CIDM tackles these problems through several innovative components, including a concept consolidation loss, elastic weight aggregation, and a context-controllable synthesis strategy. The paper provides experiments on various customization tasks to demonstrate the effectiveness of their approach compared to existing methods.

**Strengths:**

1. The paper addresses a practical problem in custom text-to-image generation, namely the ability to continually learn new concepts without forgetting old ones.
2. The ablation studies provide insights into the contribution of each proposed component, strengthening the paper's technical depth.
3. The approach is model-agnostic and has been tested on different backbone architectures (SD-1.5 and SDXL), showing its broad applicability.

**Weaknesses:**

1.  The symbol definitions are too complex which are hard to follow.
2. The paper introduces several new hyperparameters. How sensitive is the model to these?
3. How well does the model perform across different application domains and with varied types of personalized concepts?

**Questions:**

See weakness.

**Limitations:**

Yes

---

> ### Author Rebuttal · Authors · 2024-08-06
>
> Q1: The symbol definitions are too complex which are hard to follow.
>
> A1: Thanks for your valuable comment. We will carefully polish the symbol definitions and introduce a table about notation definitions to better understand this paper in the final revision process.
>
> Q2: This paper introduces several new hyperparameters. How sensitive is the model to these?
>
> A2: Thanks for your insigntful comment. We employ Stable Diffusion (SD-1.5) [38] as the pretrained model and use image alignment (IA) as metric to investigate the effect of the hyperparameters {$\gamma_1, \gamma_2, r$} on the performance of our model. When setting the rank $r=4$ of LoRAs, we analyze the effect of the hyperparameters {$\gamma_1, \gamma_2$} over a range of {0.01, 0.1, 1, 10}. Moreover, we introduce the hyperparameter experiments for $r$ within the range {2, 4, 6, 8}, when setting the balancing weights $\gamma_1=0.1, \gamma_2=1.0$. From the following results, we observe that our model has stable performance over a wide range of hyperparameters {$\gamma_1, \gamma_2, r$}. Furthermore, our model is not sensitive to different hyperparameter selections, making it effective and robust for learning continuous text-guided concept customization tasks.
>
>
> | $\gamma_1$ \ $\gamma_2$ | 0.01| 0.1 | 1.0 | 10  |
> |:-:                      |:-:  |:-:  |:-:  |:-:  |
> | **0.01**                   | 77.3|77.7 |77.9 |77.5 |
> | **0.1**                     | 77.7|77.9 | 78.0|77.6 |
> | **1.0**                     | 77.5|77.8 | 77.9|77.9 |
> | **10**                      | 77.8|77.6 | 77.4| 76.6|
> ---
>
> | rank     | 2   | 4   | 6   | 8   |
> |:-:       |:-:  |:-:  |:-:  |:-:  |
> | **IA (%)**   | 76.2|78.0 |77.8 | 77.9|
> ---
>
> Q3: How well does the model perform across different application domains and with varied types of personalized concepts?
>
> A3: Thanks for your constructive comments. Inspired by [10][46], we construct a new challenging concept-incremental learning (CIL) dataset including ten continuous text-guided concept customization tasks to verify the effectiveness of our model under the CIFC setting. This dataset has varied types of personalized concepts, which are collected from different application domains. Moreover, different concepts have significant differences. Specifically, as shown in Fig. 8, seven tasks have different object concepts (i.e., V2 duck toy, V4 backpack, V5 teddy bear) and pet concepts (i.e., V1 dog, V3 cat, V7 dog and V9 cat) from [40, 22]. Besides, the remaining three tasks have different style concepts (i.e., V6, V8 and V10 styles) collected from website. Considering the practicality of the CIFC setting, we introduce some semantically similar concepts (e.g., V1 and V7 dogs, V3 and V9 cats), making the CIL dataset more challenging under the CIFC setting. Particularly, all comparison experiments of this paper are conducted on this challenging dataset. As shown in Tabs. 1-2 and Figs. 2-5 of the submitted manuscript, our model achieves better performance than other state-of-the-art (SOTA) comparison methods in resolving the practical concept-incremental flexible customization (CIFC) problem, under both qualitative and quantitative evaluations.

---

> ### Comment · Reviewer_SpKd · 2024-08-13
>
> Thanks for the authors' responses. Some of my concerns have been solved.

---

> > ### Author Response · Authors · 2024-08-13
> > **Thank you for your valuable comments!**
> >
> > Dear Reviewer SpKd,
> >
> > We are pleased that we have addressed some of your concerns. We are very grateful to you for dedicating your time and effort to evaluating our paper.
> >
> > Best regards,
> >
> > Authors of Paper #1919

---

### Official Review · Reviewer_YpEo · 2024-07-10

**Soundness:** 4
**Presentation:** 3
**Contribution:** 3
**Rating:** 7
**Confidence:** 4

**Summary:**

This paper explores a novel and practically significant problem, namely how custom diffusion models can continuously learn new personalized concepts while avoiding catastrophic forgetting and concept neglect. The authors developed a  concept consolidation loss and an elastic weight aggregation module to explore task-specific knowledge and task-shared knowledge, and aggregated all low-rank weights of old concepts based on the contributions of old concepts. To address the problem of concept neglect, the authors propose a context-controllable synthesis strategy that can utilize expressive regional features and noise estimation to control the generated contexts. Experimental results confirm the effectiveness of the proposed method.

**Strengths:**

1. The continual learning of generative models is an interesting and meaningful task. How to mitigate catastrophic forgetting and concept neglect in concept-incremental flexible customization scenario remains insufficiently explored.
2. The method proposed by the authors effectively mitigates the challenges of catastrophic forgetting and concept neglect, with detailed descriptions and comparisons in both methodology and analysis.
3. The writing is good and easy to read.

**Weaknesses:**

1. It would be better to mention the comparison methods of experiments in the related work，such as LoRA-M, LoRA-C, CLoRA and L2DM, and emphasize the differences between them and the proposed method.
2. This paper attempts to address two challenges in the Concept-Incremental text-to-image Diffusion Model: catastrophic forgetting and concept neglect. However, from the text, I did not understand the relationship between continual learning and concept neglect. I would like to know if concept neglect is a challenge introduced by continual learning, and what impact continual learning has on original concept neglect?
3. The baseline choices for continual learning such as EWC and LWF are too outdated. Why not choose new and more effective continual learning algorithms? Intuitively, more advanced algorithms should be able to better mitigate catastrophic forgetting.
4. Due to the past focus of continual learning mainly on classification tasks, I hope to see more discussion on the background of continual generation models. For example, in this setting, how severe are catastrophic forgetting and concept neglect specifically, and whether some more objective evaluation criteria can be introduced.
5. How much do different weighting schemes between LoRAs affect the quality of generation, compared to some intuitive weighting baselines?
6. Does the value of the rank in LoRA and its placement in the model (e.g., which layer) have an impact on the results?

**Questions:**

Please check the above concerns in the weakness points. And If my concerns are resolved, I will consider increasing my score.

**Limitations:**

See the weaknesses above.

---

> ### Author Rebuttal · Authors · 2024-08-06
>
> Q1: It would be better to mention the comparison methods of experiments in the related work, such as LoRA-M, LoRA-C, CLoRA and L2DM, and emphasize the differences between them and the proposed method.
>
> A1: Thanks for your insightful comment. We will carefully polish the related work in the finial revision. The differences between our model and other methods are as follows:
>
> 1) LoRA-M [64] equally amalgamates all LoRA layers to retrain diffusion model, and LoRA-C [64] explores contributions of different LoRA layers for multi-concept composition. When learning new concepts continually, they may experience signifcant loss of attributes on old concepts (i.e., catastrophic forgetting) for versatile customization. Different from them, we devise a concept consolidation loss to mitigate forgetting of old concepts by exploring task-specifc/task-shared knowledge.
>
> 2) CLoRA [43] proposes a self-regularized low-rank adaption to continually learn new concepts, while L2DM [46] builds a long-term memory bank to reconstruct old concepts. They cannot control synthesized contexts according to user conditions and suffer from concept neglect in multi-concept composition. However, our model proposes a context-controllable synthesis to tackle concept neglect, and effectively learns new concepts continuously for versatile customization.
>
> Q2: I did not understand the relationship between continual learning and concept neglect. I would like to know if concept neglect is a challenge introduced by continual learning, and what impact continual learning has on original concept neglect?
>
> A2: As introduced in [46][64], concept neglect is a common challenge when performing multi-concept composition according to user-provided conditions, rather than being directly introduced by continual learning. However, if latent diffusion models aim to consecutively synthesize a sequence of new concepts under the CIFC setting, the catastrophic forgetting of old concepts can significantly exacerbate the degree of concept neglect when composing multiple personalized concepts.
>
> Q3: The baseline choices for continual learning such as EWC and LWF are too outdated. Why not choose new and more effective continual learning algorithms? Intuitively, more advanced algorithms should be able to better mitigate catastrophic forgetting.
>
> A3: To further verify the effectiveness of our model, we use a more advanced continual learning algorithm, InfLoRA Refs[1], published in CVPR2024, for performance comparison. As shown in the following results, our model substantially surpasses InfLoRA Refs[1] by a large margin. Note that CLoRA [43] and L2DM [46] are the two most advanced continual generation models, which are highly relevant to this paper. Particularly, our model achieves better performance than these SOTA methods.
>
> | Methods               |IA (SD-1.5) |
> |-                      |:-:    |
> | Finetuning            | 73.7  |
> | EWC [19]              | 75.9  |
> | LWF [25]              | 74.1  |
> | CLoRA [43]            | 76.9  |
> | L2DM [46]             | 76.1  |
> | InfLoRA Refs[1]       | 76.2  |
> | **CIDM (Ours)**       | **78.0** |
>
> Refs[1]  InfLoRA: Interference-Free Low-Rank Adaptation for Continual Learning, CVPR 2024
>
> Q4:	How severe are catastrophic forgetting and concept neglect, and whether some more objective evaluation criteria can be introduced.
>
> A4: Following L2DM [46], we introduce a new evaluation criteria: Task Forgetting Rate of Image Alignment (TFR-IA), to evaluate the degree of catastrophic forgetting and concept neglect. For the $l$-th concept, its TFR-IA is computed by $\frac{1}{g-1}\sum_{i=1}^{g-1} I_{i,l} - I_{g,l}$, where $I_{i,l}$ and $I_{g,l}$ denote the image alignment (IA) of the $l$-th concept evaluated at the $i$-th and $g$-th tasks. Our model outperforms other SOTA methods in terms of the TFR-IA metric. It verifies that our model has the least forgetting and concept neglect, making it more effective for continual generation.
>
> | Methods               | TFR-IA (%) |
> |-                      |:-:    |
> | EWC [19]              | 2.48  |
> | LoRA-M [64]           | 4.42  |
> | LoRA-C [64]           | 4.30  |
> | CLoRA [43]            | 1.56  |
> | L2DM [46]             | 1.37  |
> | InfLoRA Refs[1]       | 1.39  |
> | **CIDM (Ours)**       | **1.22** |
> ---
>
> Q5:	How much do different weighting schemes between LoRAs affect the quality of generation, compared to some intuitive weighting baselines?
>
> A5: As shown in Tab. 1 and Figs. 2-3 of the submitted manuscript, we introduce comparisons between our model and some intuitive LoRAs weighting baselines such as LoRA-M [64] and LoRA-C [64]. Specifically, LoRA-M [64] equally amalgamates all LoRA layers to retrain diffusion model, and LoRA-C [64] explores contributions of different LoRA layers for multi-concept composition. In Tab. 1 and Figs. 2-3, LoRA-M [64] and LoRA-C [64] experience signifcant loss of attributes on old concepts (i.e., catastrophic forgetting) for versatile customization. Moreover, our model significantly outperforms them in effectively resolving catastrophic forgetting.
>
> Q6:	Does the value of the rank in LoRA and its placement in the model (e.g., which layer) have an impact on the results?
>
> A6: Thanks for your comment.
>
> 1) Following [10], we set the value of rank to 4 in this paper, and use Stable Diffusion (SD-1.5) [38] to analyze its effect on performance (IA) by setting r={2, 4, 6, 8}. As shown in the following results, the value of rank ($r\leq 8$) has negligible impacts on the performance. Note that larger rank can increase parameters and memory costs, which violates practicality of the CIFC setting.
>
> 2) For LoRA placement, each cross-attention layer in the diffusion models has its own LoRA layer.
>
> | rank     | 2   | 4   | 6   | 8   |
> |:-:       |:-:  |:-:  |:-:  |:-:  |
> | IA (%)   | 76.2|78.0 |77.8 | 77.9|
> ---

---

> > ### Comment · Reviewer_YpEo · 2024-08-08
> >
> > Thanks for the author's response and I have another question. In the open source community of diffusion models, the generation of multi-concept images can be achieved through multi-lora composition and some regional control techniques. How can we further understand the application of continual learning in the generation community and the advantages over multi-lora composition, for example, [1][2]?
> >
> > [1] Multi-LoRA Composition for Image Generation
> >
> > [2] Mix-of-Show: Decentralized Low-Rank Adaptation for Multi-Concept Customization of Diffusion Models

---

> ### Author Response · Authors · 2024-08-10
> **Thank you for your follow-up comments!**
>
> Thank you for your follow-up comments!
>
> **1)** As for the application of continual learning in the generation community, our proposed model can help users continually synthesize a series of new personalized concepts for versatile customization based on their provided conditions, while addressing catastrophic forgetting of old personalized concepts. In contrast, existing state-of-the-art (SOTA) diffusion models in the generation community may experience significant loss of individual attributes in old personalized concepts (i.e., catastrophic forgetting) during versatile customization when continually learning new personalized concepts. The experiment results shown in Tabs. 1-2 and Figs. 2-5 of the submitted manuscript also illustrate the effectiveness of our model in addressing catastrophic forgetting and continually learning new concepts, compared to other state-of-the-art (SOTA) methods.
>
> **2) Advantages of Our model Over Multi-LoRA Composition:**
>
> **a)** Compared with SOTA multi-LoRA composition methods such as LoRA-M [64] and LoRA-C [64], our proposed elastic weight aggregation module utilizes learnable layer-wise concept tokens to merge all low-rank weights of old personalized concepts, based on their contributions to versatile concept customization. This module ensures that our model is more effective to resolve longer task sequences for versatile concept customization than other SOTA multi-LoRA composition methods such as LoRA-M [64] and LoRA-C [64]. As shown in the following results,  we use a challenging benchmark dataset (i.e., celebrity faces) to construct thirty continuous text-guided concept customization tasks. Compared with our model, current multi-LoRA composition methods (LoRA-M [64] and LoRA-C [64]) suffer from significant performance degradation, due to the catastrophic forgetting caused by the composition of longer sequence tasks.
>
> | Methods               | | Parameters (M) | | Memeory Costs (M) | |   IA (SD-1.5)       |
> |-                      |-|:-:             |-|:-:                |-|:-:             |
> | EWC [19]              | |    0.80        | |  6.87              | | 72.7$\pm$0.14  |
> | LoRA-M [64]           | |    0.40        | |  1.72             | | 72.0$\pm$0.17  |
> | LoRA-C [64]           | |    0.40        | |  1.72             | | 72.3$\pm$0.08  |
> | CLoRA [43]            | |    0.40        | |  1.72             | | 74.1$\pm$0.13  |
> | L2DM [46]             | |    0.80        | |  3.43             | |73.4$\pm$0.11  |
> | **CIDM (Ours)**       | |    0.41        | |  1.74             | | **75.7$\pm$0.09**  |
> ---
>
> **b)** Mix-of-Show [10] proposes a gradient fusion strategy to train a composed LoRA weight that mimics the predictions of individual LoRAs. However, it requires retraining the entire diffusion model and CLIP when learning new personalized concepts continually, which can substantially increase computational complexity and memory cost. As shown in the following results, compared with our model, Mix-of-Show [10] requires larger training parameters and memory costs to learn a new concept customization task, making it impractical to handle longer task sequences under the CIFC setting. Moreover, to ensure fair comparisons with existing methods, we apply the regional control techniques proposed in Mix-of-Show [10] to all comparison methods. The qualitative comparisons in Figs. 3 and 12 of the submitted manuscript demonstrate the superior performance of our model in multi-concept customization when continually learning new personalized concepts under the CIFC setting.
>
> | Methods               | | Parameters (M) | | Memeory Costs (M) |
> |-                      |-|:-:             |-|:-:                |
> | Mix-of-Show [10]      | |     983.65   | |     3930.46         |
> | **CIDM (Ours)**       | |         **0.41**   | |     **1.74**          |
> ---
>
> Refs:
>
> [10] Mix-of-Show: Decentralized Low-Rank Adaptation for Multi-Concept Customization of Diffusion Models.
>
> [64] Multi-LoRA Composition for Image Generation.

---

> > ### Comment · Reviewer_YpEo · 2024-08-10
> >
> > Thank you for your clear response. I have no further questions.

---

> > > ### Author Response · Authors · 2024-08-10
> > > **Thank you for your generous support!**
> > >
> > > Dear Reviewer YpEo,
> > >
> > > Glad to hear that your concerns have been addressed well. Thank you for your great efforts in reviewing and for the good questions.
> > >
> > > Best regards,
> > >
> > > Authors of Paper #1919

---

### Official Review · Reviewer_5EBF · 2024-07-12

**Soundness:** 3
**Presentation:** 3
**Contribution:** 3
**Rating:** 7
**Confidence:** 4

**Summary:**

This paper introduces the Concept-Incremental text-to-image Diffusion Model (CIDM), which addresses the Concept-Incremental Flexible Customization (CIFC) problem. This approach represents one of the first explorations into learning new customization tasks incrementally, effectively navigating the dual challenges of catastrophic forgetting and concept neglect. To combat catastrophic forgetting, the paper proposes a new concept consolidation loss coupled with an elastic weight aggregation module, which together helps preserve knowledge of previously learned personalized concepts. Additionally, the development of a novel context-controllable synthesis strategy specifically addresses the challenge of concept neglect, ensuring that new concepts are integrated without overshadowing existing knowledge. The effectiveness of the CIDM is validated through comprehensive experiments across a range of text-to-image generation tasks, demonstrating its capability to learn new customization tasks consecutively with notable efficacy.

**Strengths:**

1. The paper commendably defines the Concept-Incremental Flexible Customization problem, setting a precedent for addressing this emerging challenge within the field. CIFC is a practical issue crucial for continuously synthesizing personalized concepts based on user preferences in real-world applications.

2. The novel model introduced by the authors incorporates a range of innovative functions and algorithms, which are thoroughly explained and convincingly motivated. The concept consolidation loss and the elastic weight aggregation module are noteworthy among these, paired with a context-controllable synthesis strategy. These elements collectively aim to mitigate issues of catastrophic forgetting and concept neglect effectively.

3. The experimental section of the paper is detailed, providing a broad comparison of the model’s capabilities across tasks such as multi-concept generation, style transfer, and image editing.

**Weaknesses:**

1. A critical concern arises regarding the scalability of the model, specifically the accumulated memory load after learning each task. This is particularly pertinent in continual learning scenarios where efficiency is crucial.

2. The paper lacks detailed implementation information on critical tasks such as custom image editing and style transfer. Providing these details would significantly enhance the clarity and replicability of the research.

3. The concept of balancing task-specific and task-shared knowledge is intriguing and seems to be central to effectively addressing the CIFC problem. To substantiate the claims of its effectiveness, the authors should include ablation studies that isolate and quantify the impact of these components. Additionally, visualizations comparing the results with and without these strategies would provide a clearer, more direct illustration of their value.

**Questions:**

1. I recommend the authors include details on memory consumption or the number of stored parameters after each task during the rebuttal process.

2. The authors should incorporate a comprehensive description of custom image editing and style transfer processes, including specific parameters and settings used, to allow for a better understanding and evaluation of the proposed methods' effectiveness in practical applications.

**Limitations:**

There are no potential negative societal impacts.

---

> ### Author Rebuttal · Authors · 2024-08-06
>
> Q1: I recommend the authors include details on memory consumption or the number of stored parameters after each task during the rebuttal process.
>
> A1: Thanks for your constructive suggestion. We conduct comparison experiments to evaluate memory consumption and training parameters of each task, when utilizing Stable Diffusion (SD-1.5) [38] as the pretrained model to learn ten continuous text-guided concept customization tasks. As shown in the following results, the training parameters and memory requirements of our model are comparable to those of LoRA-M [64], LoRA-C [64] and CLoRA [43], but are substantially lower than those of other SOTA methods. Moreover, our model achieves better performance than  existing SOTA diffusion models (see Tabs. 1-2 and Figs. 2-5). It illustrates that our model is efficient and effective to tackle the practical Concept-Incremental Flexible Customization (CIFC) problem.
>
> | Methods               | | Parameters (M) | | Memeory Costs (M) |
> |-                      |-|:-:             |-|:-:                |
> | Finetuning            | |         0.80   | |     3.43          |
> | EWC [19]              | |         0.80   | |     6.87          |
> | LWF [25]              | |         0.80   | |     4.21          |
> | LoRA-M [64]           | |         0.40   | |     1.72          |
> | LoRA-C [64]           | |         0.40   | |     1.72          |
> | CLoRA [43]            | |         0.40   | |     1.72          |
> | L2DM [46]             | |         0.80   | |     3.43          |
> | **CIDM (Ours)**       | |         0.41   | |     1.74          |
> ---
>
> Q2: The paper lacks detailed implementation information on critical tasks such as custom image editing and style transfer. Providing these details would significantly enhance the clarity and replicability of the research.
>
> A2: Thanks for your insightful comments.
>
> 1) As for custom image editing, we introduce Anydoor [3] as a plug-in for all comparison methods. Specifically, given a text prompt that contains the user's personalized concept, an initial image, and an editable bounding box, we first input the given text prompt into the latent diffusion model to perform custom synthesis. Then we embed the generated image into the editable bounding box of the initial image using Anydoor [3].
>
> 2) We introduce T2I-adapter [30] to achieve custom style transfer. Specifically, given an initial image, we utilize BLIP [23] to extract the corresponding caption (e.g., "photo of a city skyline"). Then, we add the custom concept (e.g., [V6] style) to the caption to obtain the style transfer prompt (e.g., "photo of a city skyline in [V6] style"). Finally, we perform Canny edge detection on the initial image to extract a Canny image, and incorporate it with the style transfer prompt to achieve custom synthesis via the T2I-adapter [30].
>
> Refs:
>
> [3] AnyDoor: Zero-shot Object-level Image Customization
>
> [23] BLIP:Bootstrapping LanguageImage Pre-training
>
> [30] T2I-Adapter: Learning Adapters to Dig out More Controllable Ability for Text-to-Image Diffusion Models
>
>
> Q3: The concept of balancing task-specific and task-shared knowledge is intriguing and seems to be central to effectively addressing the CIFC problem. To substantiate the claims of its effectiveness, the authors should include ablation studies that isolate and quantify the impact of these components. Additionally, visualizations comparing the results with and without these strategies would provide a clearer, more direct illustration of their value.
>
> A3: Many thanks for analyzing our model.
>
> 1) Tab. 3 in the submitted manuscript shows ablation studies of single-concept customization to analyze the effectiveness of each module in our model, where TSP and TSH denote task-specific knowledge and task-shared knowledge in the concept consolidation loss, respectively. When we remove the TSP or TSH module, our model decreases 0.5% and 0.6% in terms of image alignment (IA) metric. It verifes the effectiveness of our model to resolve the CIFC problem by exploring both task-specific and task-shared knowledge.
>
> 2) Fig. 6 in the submitted manuscript visualizes the ablation analysis of task-specific knowledge (TSP) and task-shared knowledge (TSH) from the perspective of image generation quality. These visualization results illustrates that our model can capture task-specifc information within each customization task and explore task-shared knowledge across different tasks to tackle the CIFC problem via optimizing Eq. (2).

---

> > ### Comment · Reviewer_5EBF · 2024-08-11
> >
> > Thank you for your responses! My concerns are addressed. Hence, I increase my score.

---

> ### Author Response · Authors · 2024-08-12
> **Thank you for your kind response and support.**
>
> Dear Reviewer 5EBF,
>
> We are pleased that we have addressed most of your concerns. Thanks for your insightful comments and approval of our work.
>
> Best regards,
>
> Authors of Paper #1919

---

### Official Review · Reviewer_Th4Q · 2024-07-15

**Soundness:** 3
**Presentation:** 3
**Contribution:** 3
**Rating:** 7
**Confidence:** 5

**Summary:**

This paper tackles the problem of continually adapting text-to-image diffusion customization models.  The proposed method employs a novel concept consolidation loss, elastic weight aggregation module, and context-controllable synthesis strategy. Extensive experiments demonstrate that the proposed method performs strongly compared to SOTA baseline methods.

**Strengths:**

1. The approach is innovative and presented in a principled manner.

2. The problem setting is a strong area of impact for research with real-world applications, which is important for continual learning.

3. The experiments and analysis are very extensive, and the paper presentation is of high quality.

**Weaknesses:**

1. Please include a table that compares the training, parameter, and memory costs of your method against other methods. This comparison is crucial for a comprehensive evaluation of your method.

2. Although the results are clear and detailed, they lack confidence intervals or similar statistical measures in the quantitative metrics. Running the experiments with multiple random seeds would ensure robustness and provide a more reliable assessment.

3. The dataset used is quite limited. To fully justify your setting, you should explore more diverse datasets and longer task sequences. For instance, with only 10 simple concepts, it would be feasible to store a LoRA version of Custom Diffusion or even the Custom Diffusion deltas in memory and merge them on the fly for the target task, achieving zero forgetting in a practical manner. To truly demonstrate the impact of your method and setting, beyond what can be trivially solved by adapters or state-of-the-art LoRA-based merging methods, you should consider long task sequences or challenging benchmarks such as celebrity faces.

**Questions:**

Please see my weaknesses section. I think 1 and 2 are crucial for acceptance (would raise my score to WA if done thoroughly). Part 3 is important, too, but more difficult to address in a short period of time.

**Limitations:**

Yes

---

> ### Author Rebuttal · Authors · 2024-08-06
>
> Q1: Please include a table that compares the training, parameter, and memory costs of your method against other methods. This comparison is crucial for a comprehensive evaluation of your method.
>
> A1: Thanks for your insightful comment. As shown in the following results, we use Stable Diffusion (SD-1.5) [38] to conduct comprehensive evaluations in terms of training time, parameters, and memory costs of each task, when the number of continuous concept customization tasks is ten. All comparisons of training time are conducted under two NVIDIA RTX A6000 GPUs. Our model has comparable parameters, memory costs, and training time to LoRA-M [64], LoRA-C [64] and CLoRA [43]. Additionally, the training parameters and memory requirements of our model are substantially lower than those of other SOTA methods. More importantly, our model significantly outperforms existing SOTA comparison models for both the qualitative and quantitative evaluations on versatile generation tasks (see Tabs. 1-2 and Figs. 2-5 in the submitted manuscript). We will consider introducing these comparisons about parameters and memory costs in the final revision.
>
> | Methods               | | Training Time (h) | | Parameters (M) | | Memeory Costs (M) | IA (SD-1.5) |
> |-                      |-|:-:                |-|:-:             |-|:-:                |:-:          |
> | Finetuning            | |         0.65      | |         0.80   | |     3.43          | 73.5$\pm$0.15  |
> | EWC [19]              | |          0.83     | |         0.80   | |     6.87          | 75.8$\pm$0.22  |
> | LWF [25]              | |          1.18     | |         0.80   | |     4.21          | 74.3$\pm$0.09  |
> | LoRA-M [64]           | |          0.65     | |         0.40   | |     1.72          | 74.6$\pm$0.23  |
> | LoRA-C [64]           | |           0.65    | |         0.40   | |     1.72          | 74.7$\pm$0.06  |
> | CLoRA [43]            | |          0.90     | |         0.40   | |     1.72           | 76.8$\pm$0.08  |
> | L2DM [46]             | |           1.35    | |         0.80   | |     3.43          | 76.3$\pm$0.14  |
> | **CIDM (Ours)**       | |           0.72    | |         0.41   | |     1.74          | **77.9$\pm$0.08** |
> ---
>
> Q2: Although the results are clear and detailed, they lack confidence intervals or similar statistical measures in the quantitative metrics. Running the experiments with multiple random seeds would ensure robustness and provide a more reliable assessment.
>
> A2: Thanks for your constructive comment. We use five random seeds (0, 2021, 2022, 2023, 2024) to conduct five random experiments in terms of the image alignment (IA) metric and report their average results over five random seeds for evaluation. As shown in the following results, our model achieves better performance than other SOTA methods, which verifies the robustness and reliable effectiveness of our model to tackle the CIFC problem. Moreover, we will consider presenting averaged comparison results over multiple random seeds in the finial revision.
>
> | Methods               | |   SD-1.5 [38]  | |   SDXL [33]   |
> |-                      |-|:-:             |-|:-:            |
> | Finetuning            | | 73.5$\pm$0.15  | | 71.4$\pm$0.13 |
> | EWC [19]              | | 75.8$\pm$0.22  | | 77.5$\pm$0.07 |
> | LWF [25]              | | 74.3$\pm$0.09  | | 76.6$\pm$0.10 |
> | LoRA-M [64]           | | 74.6$\pm$0.23  | | 74.2$\pm$0.09 |
> | LoRA-C [64]           | | 74.7$\pm$0.06  | | 74.6$\pm$0.14 |
> | CLoRA [43]            | | 76.8$\pm$0.08  | | 77.7$\pm$0.06 |
> | L2DM [46]             | | 76.3$\pm$0.14  | | 77.2$\pm$0.11 |
> | **CIDM (Ours)**       | | **77.9$\pm$0.08**  | | **79.6$\pm$0.07** |
> ---
>
> Q3: The dataset used is quite limited. To fully justify your setting, you should explore more diverse datasets and longer task sequences. To truly demonstrate the impact of your method and setting, you should consider long task sequences or challenging benchmarks such as celebrity faces.
>
> A3: Thanks for your invaluable comment. To further validate the effectiveness of our model in learning longer task sequences, we use a new challenging benchmark dataset (celebrity faces) to construct thirty continuous text-guided concept customization tasks. Compared to existing SOTA models such as LoRA-M [64], LoRA-C [64], CLoRA [43], and L2DM [46], this new setting on celebrity faces dataset includes longer task sequences (i.e., thirty text-guided customization tasks), and encompasses different object concepts. Therefore, it is more challenging and better suited to verify the effectiveness in addressing the CIFC problem. We use Stable Diffusion (SD-1.5) [38] to evaluate parameters, memory costs and image aligment (IA) metric over five random seeds. From the following results, we observe that our model significantly outperforms other SOTA methods to learn longer task sequences. More importantly, the training parameters and memory requirements of our model are comparable to those of LoRA-M [64], LoRA-C [64] and CLoRA [43], but are substantially lower than those of other SOTA methods. It verifies the effectiveness of our model to learn longer task sequences on the new challenging dataset.
>
> | Methods               | | Parameters (M) | | Memeory Costs (M) | |   IA (%)       |
> |-                      |-|:-:             |-|:-:                |-|:-:             |
> | EWC [19]              | |    0.80        | |  6.87              | | 72.7$\pm$0.14  |
> | LoRA-M [64]           | |    0.40        | |  1.72             | | 72.0$\pm$0.17  |
> | LoRA-C [64]           | |    0.40        | |  1.72             | | 72.3$\pm$0.08  |
> | CLoRA [43]            | |    0.40        | |  1.72             | | 74.1$\pm$0.13  |
> | L2DM [46]             | |    0.80        | |  3.43             | |73.4$\pm$0.11  |
> | **CIDM (Ours)**       | |    0.41        | |  1.74             | | **75.7$\pm$0.09**  |

---

> > ### Comment · Reviewer_Th4Q · 2024-08-13
> >
> > I appreciate the authors' hard work and have raised my score. I hope this paper will be accepted.

---

> > > ### Author Response · Authors · 2024-08-14
> > > **Thanks for your constructive comments.**
> > >
> > > Dear Reviewer Th4Q:
> > >
> > > Thank you for supporting our work. We sincerely appreciate your insightful comments that help improve our paper. We will take them into account when making the final revisions.
> > >
> > > Best regards,
> > >
> > > Authors of Paper #1919

---

### Author Rebuttal · Authors · 2024-08-06

Dear reviewers and area chairs:

We extend our gratitude to all the reviewers and area chairs for dedicating their time and effort to evaluating our paper. We also thank the reviewers for their positive and insightful comments, which can help us improve our work.

We are encouraged that:

$\bullet$ Reviewer Th4Q and Reviewer 5EBF agree that our work is **novel** in resolving **a practical problem** named concept-incremental flexible customization (CIFC) and includes **comprehensive evaluation experiments** to validate the effectiveness of the proposed model.

$\bullet$ Reviewer YpEo thinks that our paper focuses on tackling **an interesting and meaningful task** with **extensive comparisons** in both methodology and analysis.

$\bullet$ Reviewer SpKd believes that our model addresses **a practical problem**, has **in-depth ablation studies**, and demonstrates **broad applicability** across different backbone architectures.

All reviewers recognize that our model achieves **state-of-the-art performance under comprehensive experiments**.

We have responded to each reviewer individually to address any comments. We would like to give a brief summary:

**To Reviewer Th4Q:**

1) We introduce comparison experiments on training times, parameters, and memory costs between our model and other models.

2) We conduct five random evaluation experiments and report their average results over five random seeds (0, 2021, 2022, 2023, 2024) to compare the performances.

3) We introduce longer task sequences on a new challenging dataset (celebrity faces) to further validate the effectiveness of our model.

**To Reviewer 5EBF:**

1) We provide comparisons on memory consumption and training parameters.

2) We introduce implementation details of custom image editing and style transfer.

**To Reviewer YpEo:**

1)  We clarify the significant differences between our model and other methods such as LoRA-M, LoRA-C, CLoRA, and L2DM, and explain the relationship between continual learning and concept neglect.

2) We use a more advanced continual learning algorithm, InfLoRA [1], published in CVPR2024, for performance comparison.

3) We introduce a new evaluation criteria: Task Forgetting Rate of Image Alignment (TFR-IA), to assess the degree of catastrophic forgetting and concept neglect.

4) We compare our model with several intuitive LoRA weighting baselines, such as LoRA-M [64] and LoRA-C [64], and discuss the impact of low rank in LoRAs.

Refs[1] InfLoRA: Interference-Free Low-Rank Adaptation for Continual Learning, CVPR 2024


**To Reviewer SpKd:**

1) We will carefully polish the symbol definitions and introduce a table about notation definitions in the appendix.

2) We investigate the effect of the hyperparameters {$\gamma_1, \gamma_2, r$} on the performance of our model.

3) We clarify that our model can perform well on the challenging dataset used in this paper, which contains various types of personalized concepts collected from different application domains.

We thank all reviewers and area chairs again!

Best,

Authors of Paper #1919

---

### Decision · Program_Chairs · 2024-09-25

**Decision:**

Accept (poster)

**Comment:**

This paper makes a significant contribution to the field of continual learning in generative models, with a particular focus on personalized text-to-image generation. The proposed CIDM method introduces innovative solutions to the challenges of catastrophic forgetting and concept neglect, addressing a problem of high relevance and complexity. The paper demonstrates a solid level of novelty and technical rigor, supported by a well-executed experimental evaluation. Based on the strength of the contributions, the clarity of presentation, and the comprehensive coverage of experiments, the review committee has reached a consensus to recommend this paper for acceptance.